# Vaccination decreases the risk of influenza A virus reassortment but not genetic variation in pigs

Chong Li[1]*, Marie R Culhane[1], Declan C Schroeder[1], Maxim C-J Cheeran[1], Lucina Galina Pantoja[2], Micah L Jansen[2], Montserrat Torremorell[1]*

[1]College of Veterinary Medicine, University of Minnesota, Saint Paul, United States; [2]Zoetis, Parsippany, United States

**Abstract** Although vaccination is broadly used in North American swine breeding herds, managing swine influenza is challenging primarily due to the continuous evolution of influenza A virus (IAV) and the ability of the virus to transmit among vaccinated pigs. Studies that have simultaneously assessed the impact of vaccination on the emergence of IAV reassortment and genetic variation in pigs are limited. Here, we directly sequenced 28 bronchoalveolar lavage fluid (BALF) samples collected from vaccinated and unvaccinated pigs co-infected with H1N1 and H3N2 IAV strains, and characterized 202 individual viral plaques recovered from 13 BALF samples. We identified 54 reassortant viruses that were grouped in 17 single and 16 mixed genotypes. Notably, we found that prime-boost vaccinated pigs had less reassortant viruses than nonvaccinated pigs, likely due to a reduction in the number of days pigs were co-infected with both challenge viruses. However, direct sequencing from BALF samples revealed limited impact of vaccination on viral variant frequency, evolutionary rates, and nucleotide diversity in any IAV coding regions. Overall, our results highlight the value of IAV vaccination not only at limiting virus replication in pigs but also at protecting public health by restricting the generation of novel reassortants with zoonotic and/or pandemic potential.

*For correspondence:
lixx5577@umn.edu (CL);
torr0033@umn.edu (MT)

## Editor's evaluation

Vaccines are a major influenza control strategy in swine but perform sub-optimally and are under-utilized. The manuscript describes a detailed genetic characterization of influenza virus variants in vaccinated versus unvaccinated pigs. The results indicate that viral reassortment, which is an important process yielding new strange influenza viruses of importance to man and animals, may be less common in pigs that have been vaccinated against influenza.

## Introduction

Influenza A viruses (IAV) are important pathogens in humans, birds, and pigs globally. In the United States, IAV infection causes a significant disease burden on the healthcare system and society, resulting in 12,000–61,000 human deaths plus estimated losses of $11.2 billion annually (*Centers for Disease control and Prevention, 2020*; *Putri et al., 2018*). IAV infections in pigs are also considered one of the top disease concerns for the US swine industry when influenza-induced respiratory disease severely reduces pig health and subsequently the pork producers' profitability and sustainability. Human and swine share the same IAV subtypes (*Ryt-Hansen et al., 2021*; *Vincent et al., 2014*), which have similar tangled evolutionary histories contributing to the bidirectional transmission of IAV between both species (*Anderson et al., 2021*). Pigs pose a risk for generating novel IAV strains with

**eLife digest** Swine influenza A viruses cause severe illness among pigs and financial losses on pig farms worldwide. These viruses can also infect humans and have caused deadly human pandemics in the past. Influenza A viruses are dangerous because viruses can be transferred between humans, birds and pigs. These co-infections can allow the viruses to swap genetic material. Viral genetic exchanges can result in new virus strains that are more dangerous or that can infect other types of animals more easily.

Farmers vaccinate their pigs to control the swine influenza A virus. The vaccines are regularly updated to match circulating virus strains. But the virus evolves rapidly to escape vaccine-induced immunity, and infections are common even in vaccinated pigs. Learning about how vaccination affects the evolution of influenza A viruses in pigs could help scientists prevent outbreaks on pig farms and avoid spillover pandemics in humans.

Li et al. show that influenza A viruses are less likely to swap genetic material in vaccinated and boosted pigs than in unvaccinated animals. In the experiments, Li et al. collected swine influenza A samples from the lungs of pigs that had received different vaccination protocols. Next, Li et al. used next-generation sequencing to identify new mutations in the virus or genetic swaps among different strains. In pigs infected with both the H1N1 and H3N2 strains of influenza, the two viruses began trading genes within a week. But less genetic mixing occurred in vaccinated and boosted pigs because they spent less time infected with both viruses than in unvaccinated pigs. The vaccination status of the pig did not have much effect on how many new mutations occurred in the viruses.

The experiments show that vaccinating and boosting pigs against influenza A viruses may protect against genetic swapping among influenza viruses. If future studies on pig farms confirm the results, the information gleaned from the study could help scientists improve farm vaccine protocols to further reduce influenza risks to animals and people.

zoonotic and pandemic potential, which represents an unpredictable threat to both the swine industry and global public health.

IAV exhibits an extraordinary ability for cross-species transmission and immune evasion by continually expanding its genetic diversity through antigenic drift and shift. This genetic diversity permits the rapid evolution of IAV, maximizing the virus's opportunity to remain viable following significant changes in the environment and making the use of vaccines extremely difficult for disease prevention (*Martin and Brooke, 2019*; *Sandbulte et al., 2015*; *Vincent et al., 2017*). Therefore, minimizing IAV diversity should be a key strategy for One Health purposes to efficiently control IAV transmission between humans and pigs (*Centers for Disease Control and Prevention, 2017*). As swine are susceptible to avian, human, and swine-origin influenza virus, IAV introduction from multiple hosts immensely enriches the genetic pool of swine IAV and is responsible for the emergence of distinct H1 and H3 IAV lineages during the last 20 years in pigs (*Lewis et al., 2016*; *Ma et al., 2009*). The distribution of IAV receptors in the swine respiratory tract also promotes IAV co-infections with strains from various hosts and facilitates virus reassortment that may result in new viruses (*Van Reeth, 2007*). A mathematical model estimated an overall probability of a co-infection event in a farrow-to-finish pig farm, possibly leading to reassortment, to be 16.8% (*Cador et al., 2017*). This estimate seems reasonable given that the recurrent and co-circulation of distinct IAV subtypes within swine herds is quite common and that over 74 different H1 genotypes have been detected in the US pig population alone from 2009 to 2016 (*Gao et al., 2017*; *Nirmala et al., 2021*; *Rose et al., 2013*). With such diverse viral populations, swine should be considered one of the potential sources for novel IAV variants of zoonotic and pandemic infections. The 2009 H1N1 pandemic virus (H1N1pdm09), a reassortant virus that originated in pigs, contained gene segments of avian, swine, and human influenza viruses that led to the first influenza pandemic of the 21st century (*Garten et al., 2009*). Within the first year of circulation, between 151,700 and 575,400 people died worldwide due to the 2009 H1N1 virus infection (*Centers for Disease Control and Prevention, 2019*). Due to the continued evolution, the H1N1pdm09 virus has generated a complex hemagglutinin (HA) clade system (https://nextstrain.org/flu/seasonal/h1n1pdm/ha/12y; *Hadfield et al., 2018*). Moreover, the H1N1pdm09 virus has further reassorted with endemic IAV strains in pigs and has generated reassortants with distinct genetic

constellations in many countries (*Charoenvisal et al., 2013*; *Howard et al., 2011*; *Lam et al., 2011*; *Rajão et al., 2017*; *Wong et al., 2012*). Some reassortants of specific genotypes have already become the predominant circulating strains in swine populations and have caused fatal infections in people in contact with pigs (*Resende et al., 2017*; *Sun et al., 2020*).

Vaccination against IAV is the primary measure to prevent influenza in pigs. In the United States, influenza vaccination is implemented in over 80% of large breeding herds and more than 95% of breeding females in these herds receive at least two doses of vaccines before their first farrowing (*USDA, 2016*). Even though live-attenuated influenza vaccine (LAIV) and RNA-based influenza vaccines have recently been licensed in the United States, whole-cell inactivated vaccines with oil-based adjuvants are still the most common vaccine types in pigs (*Aphis, 2020*; *Van Reeth and Ma, 2013*). To maximize cross-protective immunity against multiple viruses circulating in pig herds, most commercial vaccines contain multiple antigenically distinct IAV strains. However, IAV breakthrough infections are common in pigs as the virus evolves rapidly to escape host immunity (*Murcia et al., 2012*), resulting in the circulation of the virus in vaccinated herds (*Chamba Pardo et al., 2021*). Besides, the mismatch between vaccine and field strains raises concerns about the vaccine-associated enhanced respiratory disease (VAERD), which has been reported in pigs vaccinated with a univalent whole inactivated vaccine (WIV) and experimentally challenged with a heterologous virus (*Khurana et al., 2013*; *Rajão et al., 2016*). VAERD is attributed to non-neutralizing antibodies induced by vaccines failing to neutralize the virus and instead exacerbate virus replication and disease enhancement (*Crowe, 2013*; *Khurana et al., 2013*). Therefore, understanding how immunity induced by swine IAV vaccination shapes within-host virus evolution in pigs is key to controlling the disease and the emergence of novel antigenic variants. Previous studies have characterized the IAV mutational spectra within naïve and vaccinated pigs and other mammals to simulate the impact of immune pressure on within-host diversity (*Debbink et al., 2017*; *Hoelzer et al., 2010*; *McCrone et al., 2018*; *Murcia et al., 2013*; *Murcia et al., 2012*). However, in vivo studies that explore how vaccination impacts reassortment between multiple subtypes of IAVs in pigs are lacking. Besides, most of the knowledge on IAV within-host diversity in pigs so far is based on samples taken from nasal cavities and studies that quantified the IAV within-host variation in pig lungs are lacking (*Illingworth et al., 2014*; *Murcia et al., 2012*). Considering that tissue tropism plays a vital role in IAV dissemination along the swine respiratory tract (*Zhang et al., 2018*), with the pig lungs harboring IAV populations with the most extensive genomic variations, the effect of immune pressure on IAV within-host diversity may differ by anatomical location (*Takayama et al., 2021*; *Xue et al., 2018*; *Zhang et al., 2018*). Therefore, in vivo studies are needed to provide an integrated picture of how vaccine-induced immunity affects IAV evolutionary trajectories occurring in the swine lower respiratory tract by evaluating the extent of IAV reassortment and mutational spectra taking place concurrently in naïve and vaccinated pigs.

We previously published a vaccine-challenge study assessing IAV infections in pigs vaccinated with five distinct prime-boost vaccine combinations after simultaneous infection with both an H1N1 and an H3N2 IAV strain using a seeder pig model (*Li et al., 2020*). The bronchoalveolar lavage fluid (BALF) samples obtained from the aforementioned study enabled us to evaluate, in this study, how IAV reassorts and mutates in the swine lower respiratory tract under immune pressure. We hypothesized that the dual-subtype IAV co-infection model better represents the conditions of IAV co-infection encountered in the field and that the findings could contribute to the body of knowledge of within-host virus evolution in pigs (*Nirmala et al., 2021*). Here, we performed next-generation sequencing directly on BALF samples and IAV plaques purified from the BALF samples to identify the virus mutations and reassortment that happened in swine lungs.

## Results

### Specimen collection and background information

The BALF specimens utilized in this study originated from a previously published vaccine-challenge study (*Li et al., 2020*). We used a co-infection challenge model that attempted to simulate more realistic field settings by commingling two infected seeder pigs, one inoculated with an H1N1 (A/swine/Minnesota/PAH-618/2011) and the other with an H3N2 (A/swine/Minnesota/080470/2015) influenza virus, with 10 other contact pigs in the same room to attempt simultaneous infection to both strains using a natural transmission route (*Figure 1*). A total of 14 seeder pigs and 70 in-contact pigs were

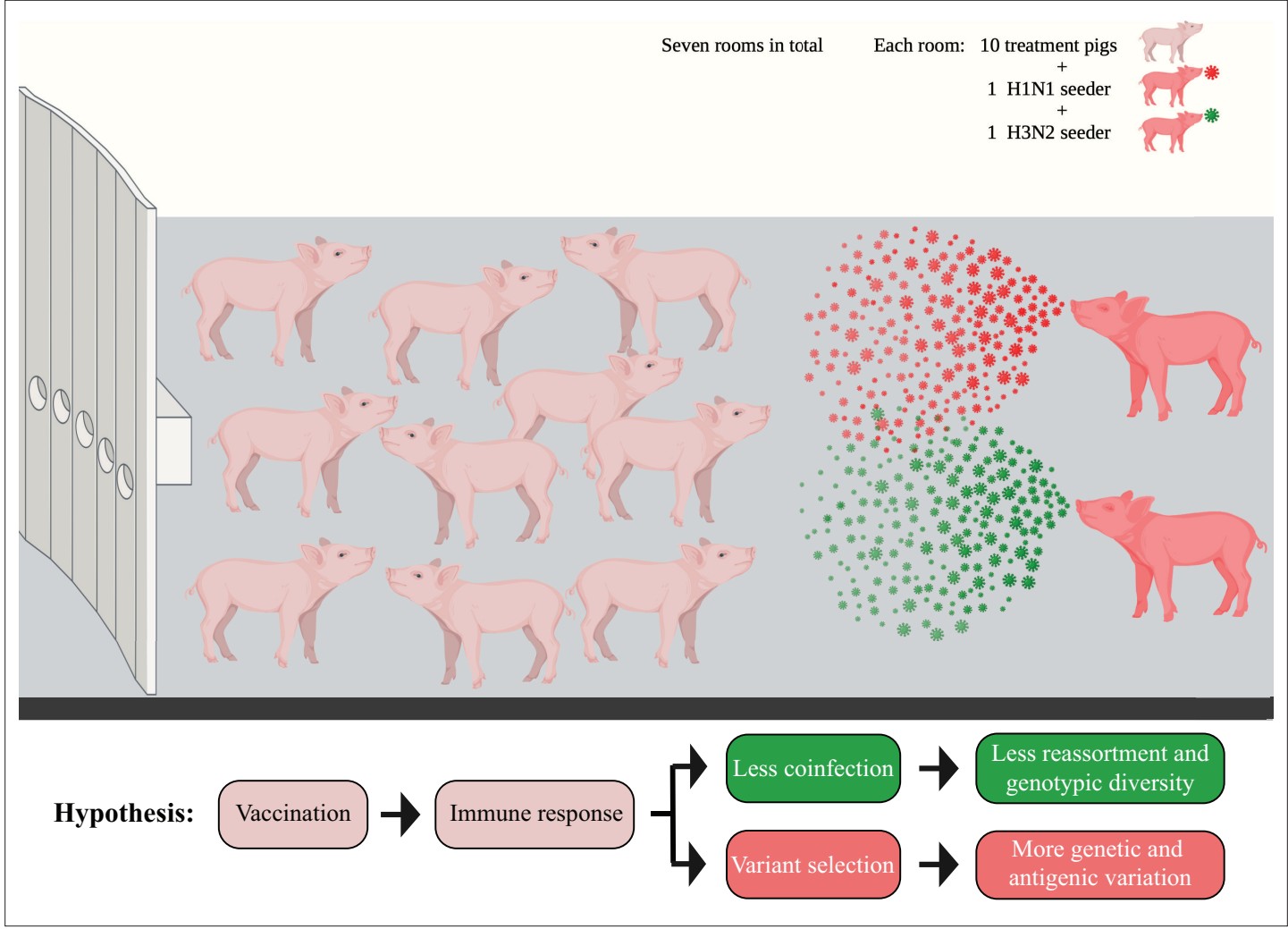

**Figure 1.** Diagram showing the seeder pig infection model. Fourteen naïve pigs (seeder pigs) were either challenged using an H1N1 or an H3N2 virus and evenly distributed in pairs into seven rooms at approximately 48 hr post inoculation. Two seeder pigs (one H1N1 seeder and one H3N2 seeder) served as the infection sources and commingled with 10 treatment pigs within each room. The treatment pigs had been vaccinated using different vaccine combinations of a commercial multivalent whole inactivated vaccine (COM), an autogenous multivalent whole inactivated vaccine (AUT), or a bivalent live-attenuated vaccine (LAIV). Bronchoalveolar lavage fluid (BALF) samples from all treatment pigs were collected at necropsy (7 days post contact [DPC] with the seeder pigs). *Figure 1* has been adapted from Figure 1 from *Li et al., 2020* created by BioRender.com.

distributed into seven rooms. The two challenge viruses contained clade 1A 3.3.3 gamma (H1N1) or clade 3.2010.1 human-like (H3N2) HA, which represent the major circulating strains in the US pig industry (*Aphis, 2022*). Three licensed IAV vaccines were used in the study, including a commercial (COM) quadrivalent WIV, an autogenous (AUT) trivalent WIV, or a bivalent LAIV. The protective effect of four different WIV combinations (including COM/COM, AUT/AUT, AUT/COM, and COM/AUT) and positive control treatment (NO VAC/CHALL) was evaluated in 50 pigs that had been distributed evenly in groups of 2 pigs per treatment to each of five rooms. Another 20 pigs received two different administrations of the LAIV (LAIV/COM and LAIV/NONE), and pigs were distributed evenly in groups of 5 per treatment to each of the two rooms.

The four WIV administrations significantly decreased virus shedding, and details of the results can be found in *Li et al., 2020*. Briefly, there were significant differences in IAV average real-time PCR (rRT-PCR) Ct values or virus titers (TCID50/ml) between the NO VAC/CHA group and the WIV treatment groups in nasal swabs and BALF samples at most time points. There were no differences in shedding between the four WIV groups except for the COM/COM group that had significantly lower Ct values in lungs compared to AUT/COM and AUT/AUT groups. In addition, delivering a COM vaccine boost

**Table 1.** Number of bronchoalveolar lavage fluid (BALF) samples available for direct sequencing or plaque purification from each group.

| Group | Vaccination protocol ‡ | Prime | Boost | No. samples sequenced/ total | Average Ct value (range) | Average coverage on H1N1 consensus genome (SD ) * | Average coverage on H3N2 consensus genome (SD) | No. samples assayed to obtain purified plaques/total † |
|---|---|---|---|---|---|---|---|---|
| | COM/COM | COM | COM | 5/10 pigs | 22.47 (17.83–27.58) | 8397 (722) | 12,897 (1329) | 3/4 pigs |
| | AUT/AUT | AUT | AUT | 1/10 pigs | 26.12 (NA ) | 3192 (NA) | 7769 (NA) | 0/4 pigs |
| | COM/AUT | COM | AUT | 2/10 pigs | 31.10 (30.88–31.31) | 18,267 (NA) | 5784 (6318) | 0/4 pigs |
| | LAIV/COM | LAIV | COM | 4/10 pigs | 26.74 (20.30–30.63) | 415 (NA) | 18,209 (12333) | 2/5 pigs |
| PRIME BOOST | In total | | | 12/40 pigs | 25.64 (17.83–31.31) | 7733 (6829) | 13,069 (8611) | 5/17 pigs |
| SINGLE LAIV | LAIV/NONE | LAIV | Saline | 9/10 pigs | 22.84 (17.76–29.84) | 9322 (4090) | 9009 (5513) | 4/5 pigs |
| NO VAC | NO VAC/ CHA | Saline | Saline | 7/10 pigs | 21.17 (14.97–27.73) | 7699 (4406) | 9775 (1057) | 4/4 pigs |
| In total | | | | 28/60 pigs | 23.62 (14.97–31.31) | 8306 (4801) | 11,281 (6972) | 13/26 pigs |

Ct, cycle threshold; NA, not applicable; SD, standard deviation.

*The genome coverage was computed as the mean depth of the trimmed gene reads covered on the sequenced influenza (IAV) genomes across all sequenced samples. The gene segments whose coverage was below 100 reads were discarded for coverage calculation and SNV identification.

†The samples that yield the purified IAV plaques were also directly sequenced by the next-generation sequencing platform.

‡The vaccines used in this study include a commercial multivalent whole inactivated vaccine (COM), an autogenous multivalent whole inactivated vaccine (AUT), or a bivalent live-attenuated vaccine (LAIV).

to pigs following a single LAIV administration provided better protection and significantly decreased virus shedding in the nasal cavities and lungs. Even though we could not perform direct statistical comparisons on vaccine protection between WIV and LAIV treatment groups as they represented two independent parts of the study, similar IAV loads and shedding patterns were observed among all WIV and LAIV/COM groups and between NO VAC/CHALL and LAIV/NONE groups. The HA sequencing data of nasal swabs and BALF samples revealed that more contact pigs were infected with the H3 subtype than with the H1 IAV subtype for all treatment groups, except for the IAV infections reported in the lungs of NO VAC/CHALL and LAIV/NONE pigs (*Li et al., 2020*).

Overall, there were 70 BALF samples from the treatment pigs collected at necropsy 7 days post contact (DPC) with the seeder pigs. Also, 28 of the BALF samples were successfully sequenced and their genomes were used to identify IAV mutations (*Table 1*). To identify IAV reassortants in vaccinated and nonvaccinated pigs, we performed plaque assays on BALF samples from pigs in three rooms (two rooms containing WIV treatment pigs and one room containing LAIV treatment pigs). These rooms were selected because seeder pigs shed IAV in enough quantities to transmit it to in-contact pigs. There were 13 BALF samples that could be used for plaque isolation, yielding a total of 202 IAV plaques (*Table 1*). Based on the vaccination regime, treatment pigs were grouped into PRIME BOOST (with COM/COM, AUT/AUT, COM/AUT, and LAIV/COM pigs), SINGLE LAIV (LAIV/NONE pigs), and NO VAC (NO VAC/CHALL pigs) groups and used for further analysis. No matter for IAV mutation or reassortment identification, PRIME BOOST pigs had significantly higher specific antibody responses against both H1 and H3 challenge viruses than SINGLE LAIV and NO VAC pigs (*Supplementary file 1*). Our overall hypothesis was that vaccination helps reduce the risk of IAV co-infection in pigs and decreases the number and diversity of genotypes of reassortant viruses generated in pig lungs. At the same time, vaccine-induced immunity may drive variant selection on IAV genomes, which could affect the within-host genetic diversity and expand the antigenic diversity of IAV populations (*Figure 1*).

## Multiple genotypes identified among the new reassortant, plaque-purified viruses

To characterize and evaluate the distribution of reassortant viruses in vaccinated and nonvaccinated pigs, a total of 202 IAV plaques were isolated and whole-genome sequenced from 13 BALF samples collected at necropsy from the pigs receiving the prime-boost (PRIME BOOST), single-dose LAIV (SINGLE LAIV), and no vaccine (NO VAC) administrations (*Supplementary file 2*). A summary of the genotypes is shown in *Figure 2*. Among the 202 plaques, 148 (73.3%) were the parental virus-challenge strains (137 [67.8%] H3N2 and 11 [5.4%] H1N1). Also, 54 (26.7%) plaques were classified as IAV reassortants with 33 (16.3%) plaques distributed into 17 distinct single reassortants (R01–R17) and 21 (10.4%) plaques classified as 16 mixed reassortant genotypes (M01–M16) (*Figure 2*, *Figure 2— source data 1*). Mixed genotypes were reassortant viruses that contained complete gene sequences of both parental viruses in a given gene segment. The IAV reassortants were detected in 6 out of 13 pigs with some pigs having as few as one genotype and as many as 13 genotypes (including mixed genotypes); notably 83.3% (45/54) of reassortants originated from only three pigs (*Figure 2—figure supplement 1*). We did not detect any gene segments that originated from the LAIV strains, and in this study there were no reassortants observed between LAIV and challenge viruses.

## Vaccination decreases the number of reassortant influenza A viruses

To evaluate whether vaccination alters the occurrence of IAV reassortment in swine lungs, we compared the percentage of IAV reassortants isolated from vaccinated and nonvaccinated pigs (*Figure 3A*). Nonvaccinated (NO VAC) pigs had more reassortant viruses and more distinct genotypes, with 50.0% (37/74) of plaques being reassortants, which belong to 11 single and 14 mixed genotypes. For the remaining plaques isolated in PRIME BOOST and SINGLE LAIV pigs, 13.3% (17/128) were reassortants distributed in seven single and three mixed genotypes. We found 25.0% (13/52) of plaques isolated from the SINGLE LAIV pigs and 5.3% (4/76) of plaques in the PRIME BOOST pigs were identified as reassortants. After accounting for the unequal quantities of plaques isolated from individual pigs and the different number of pigs between treatment groups, we found that the proportion of reassortants was significantly lower in the PRIME BOOST pigs than in the NO VAC pigs (p=0.020).

To further investigate whether the occurrence of reassortant viruses increased with increasing length of time (in days) that pigs were co-infected, we evaluated the relationship between the duration of co-infection (i.e., number of days the pigs were co-infected with H1N1 and H3N2 viruses), as measured by a subtype-specific multiplex rRT-PCR in nasal swabs and BALF samples collected at 2–7 DPC and the frequency of reassortants detected in BALF at necropsy performed on 7 DPC (*Figure 3—source data 1*). We found the duration of IAV co-infections (in days) was significantly longer in NO VAC pigs than in the PRIME BOOST pigs (p=0.012, Dunn's test with Benjamini–Hochberg correction). In addition, a strong positive correlation ($R = 0.73$, p=0.0046), as evaluated by the Spearman's rank-order correlation test, was observed between the proportion of reassortant viruses isolated and the increasing duration of co-infection (*Figure 3B*).

## Virus load and genome coverage of direct sequenced BALF samples for single-nucleotide variant (SNV) identification

Virus load and sequencing coverage are considered important factors that could affect the accuracy of IAV SNV identification (*McCrone and Lauring, 2016*). The sequenced samples in this study had a high virus load with a mean Ct value of 23.62 (range 14.97–31.31) tested by IAV matrix rRT-PCR. The average Ct values of sequenced samples were not significantly different between treatment groups (p=0.13, Kruskal–Wallis rank-sum test). The mean depth across the whole available samples used for SNV identification in the H1N1 and H3N2 genomes are 8306 and 11281 reads, respectively (*Table 1*). The mean genome coverages on H1N1 and H3N2 viruses from the sequenced samples were not significantly different between treatment groups (H1N1: p=0.49, H3N2: p=0.43, Kruskal–Wallis rank-sum test).

| Genotype | PB2 | PB1 | PA | HA | NP | NA | M | NS | Reassortant | Total number |
|---|---|---|---|---|---|---|---|---|---|---|
| H3N2 | G | G | G | G | G | G | G | G | No | 137 |
| H1N1 | R | R | R | R | R | R | R | R | No | 11 |
| R01 | G | R | R | G | G | G | R | G | Yes | 3 |
| R02 | G | G | G | G | G | G | G | R | Yes | 2 |
| R03 | G | R | R | G | G | G | G | G | Yes | 4 |
| R04 | G | G | G | G | G | G | R | G | Yes | 1 |
| R05 | R | G | G | G | R | G | G | G | Yes | 1 |
| R06 | G | R | G | G | G | G | G | G | Yes | 1 |
| R07 | G | G | G | G | R | G | R | G | Yes | 1 |
| R08 | G | G | R | G | G | G | G | G | Yes | 1 |
| R09 | G | R | R | G | G | R | G | G | Yes | 1 |
| R10 | G | G | G | G | G | G | G | G | Yes | 2 |
| R11 | R | G | G | G | R | G | G | G | Yes | 1 |
| R12 | G | G | G | G | G | R | G | R | Yes | 1 |
| R13 | G | R | R | R | R | R | R | R | Yes | 9 |
| R14 | R | G | R | R | R | R | R | G | Yes | 2 |
| R15 | G | R | G | G | R | G | G | G | Yes | 1 |
| R16 | G | G | G | G | G | G | R | G | Yes | 1 |
| R17 | R | R | R | G | G | G | G | G | Yes | 1 |
| M01 | G | G | B | G | G | G | G | G | Yes | 3 |
| M02 | G | R | G | G | B | G | G | G | Yes | 1 |
| M03 | B | R | B | B | B | B | R | R | Yes | 1 |
| M04 | B | G | R | B | B | B | G | R | Yes | 2 |
| M05 | G | R | B | B | R | G | G | G | Yes | 1 |
| M06 | G | G | B | G | B | B | G | R | Yes | 2 |
| M07 | G | G | R | G | R | B | G | R | Yes | 1 |
| M08 | G | R | B | B | B | G | R | G | Yes | 1 |
| M09 | R | G | B | G | G | B | G | G | Yes | 1 |
| M10 | G | G | B | B | B | G | R | G | Yes | 1 |
| M11 | G | G | B | G | G | R | G | R | Yes | 2 |
| M12 | G | G | B | G | B | G | G | G | Yes | 1 |
| M13 | G | G | R | B | B | G | G | G | Yes | 1 |
| M14 | B | R | R | R | R | G | R | G | Yes | 1 |
| M15 | G | R | B | G | R | G | R | G | Yes | 1 |
| M16 | G | G | B | G | G | B | G | G | Yes | 1 |
| Total | | | | | | | | | | 202 |

(Legend: R = red block, G = green block, B = blue block)

**Figure 2.** Summary of genotypes detected in the influenza A virus (IAV) plaques. A total of 202 plaques were whole-genome sequenced and genotyped based on the origins of IAV gene segments in each plaque. Gene segments are shown above the columns. Red blocks represent gene segments that originate from the H1N1 virus; green blocks originate from the H3N2 virus; and blue blocks indicate that complete gene segments were detected from both viruses. The specific genotype number is indicated on the left side of each row, and the quantity of plaques that contain the corresponding gene

*Figure 2 continued on next page*

*Figure 2 continued*

constellation is shown on the right side of the row. The specific single reassortant genotype number is named after R, and the M-number indicates the specific mixed reassortant genotype number. The quantity and genotypes of influenza plaques isolated from each individual pig are displayed in *Figure 2—figure supplement 1*. The maximum likelihood trees and the assembled nucleotide sequences of isolated plaques used for constructing the trees can be found in *Figure 2—source data 1* and *Figure 2—source data 2*, respectively.

The online version of this article includes the following source data and figure supplement(s) for figure 2:

**Source data 1.** Phylogenetic analysis of influenza plaques isolated from pigs with various vaccination statuses.

**Source data 2.** Influenza consensus nucleotide sequences from isolated plaques.

**Source data 3.** Source file for *Figure 2*.

**Figure supplement 1.** Percentage of reassortant plaques by genotype detected in individual pigs.

## Vaccine-induced immunity has a limited impact on within-pig HA nucleotide variation of H1N1 and H3N2 viruses

We called a total of 380 SNVs in 218,821 sequenced H1N1 consensus nucleotides (207 nonsynonymous, 165 synonymous, 8 stop-gained) from 19 BALF samples (*Figure 4—figure supplement 1A*, *Figure 4—source data 1*). H1N1 SNVs were dominated by low-frequency variants (*Figure 4A*). About 9.5% (36/380) of SNVs existed at the consensus level (above 50% of virus population) and over 81.8% (311/380) of SNVs, which included 171 (82.6%) nonsynonymous, 132 (80.0%) synonymous, and 8 (100.0%) stop-gained SNVs, were present at less than 10% frequency. The average number

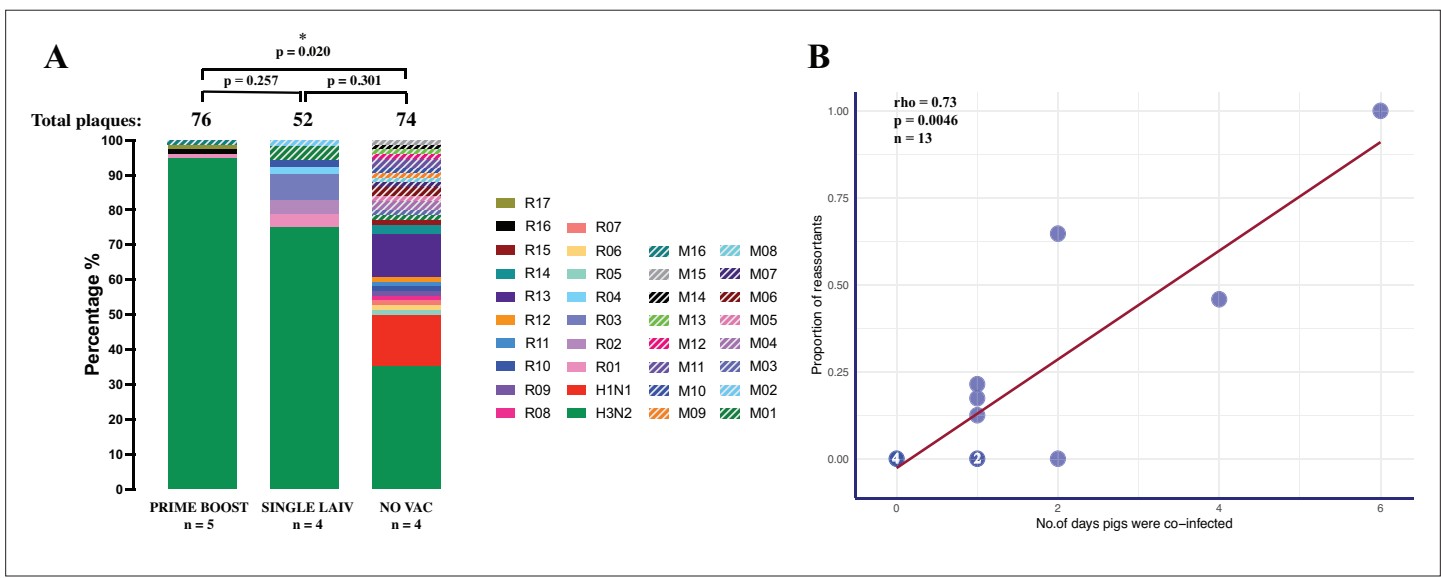

**Figure 3.** Emergence of reassortant influenza A viruses (IAV) is correlated with the number of days that pigs are co-infected with H1 and H3 viruses. (**A**) Percentage of reassortant plaques in pigs by genotype and treatment groups. Each genotype is shown in a different color. The total number of plaques for each group is shown above each bar, and the quantity of available bronchoalveolar lavage fluid (BALF) samples for each group (n) is indicated under the treatment names. Both plaques with single (R01–R17) and mixed (M01–M16) genotypes were considered as reassortant viruses. To account for the unequal number of plaques and pigs from different treatment groups, we compared the proportion of IAV reassortants by the binomial logistic regression model, allowing for overdispersion, and tested all pairwise differences between treatments using the chi-squared deviance test, and adjusted the p-values by Bonferroni–Holm method for multiple comparisons. p-value<0.05 was considered significant. (**B**) Correlation between the proportion of reassortant viruses and the number of days pigs were co-infected with H1 and H3 challenge viruses. The number in the dot represents the number of overlapping points plotted for pigs that had the same proportion of reassortants and co-infection day, and the total number of samples available for this analysis is indicated (n). The number of days pigs were co-infected is shown in *Figure 3—source data 1*, which is defined as the number of days when both H1 and H3 IAV were detected in the nasal cavities or lungs by a hemagglutinin (HA) subtype-specific multiplex rRT-PCR. Spearman's rank-order correlation test evaluated the direction and intensity of the correlation between the proportion of reassortant viruses and the number of days pigs were co-infected.

The online version of this article includes the following source data for figure 3:

**Source data 1.** Infection dynamics of H1N1 and H3N2 challenge viruses assessed by subtype specific rRT-PCR in nasal swabs and bronchoalveolar lavage fluid (BALF) samples.

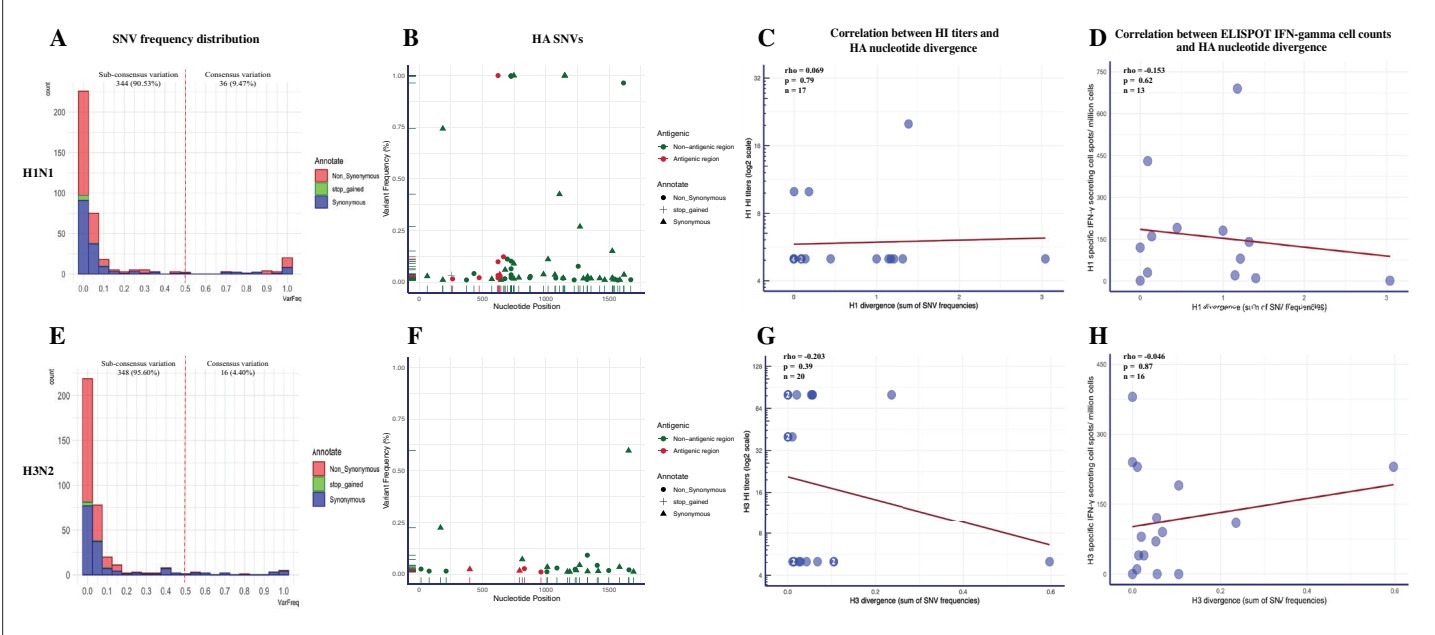

**Figure 4.** Summary of single-nucleotide variant (SNV) frequency and hemagglutinin (HA) nucleotide variations of H1N1 and H3N2 virus within pigs from different vaccination statuses. The frequency distribution of the H1N1 (A) and H3N2 (E) SNVs in pigs regardless of treatment groups. The quantity of SNVs at a given frequency interval (bin width = 0.05) is presented by a stacked histogram based on the mutation types. The SNVs with frequencies above 0.5 were considered as consensus variants. Antigenic variant identification from the total detected H1(B) and H3 (F) SNVs. The HA SNVs are shaped by their mutation types and colored based on whether they fell into the H1 antigenic regions (including the Sa, Sb, Ca1, Ca2, and Cb regions) (*Brownlee and Fodor, 2001*; *Caton et al., 1982*) or H3 antigenic regions (including the A, B, C, D, and E regions) (*Debbink et al., 2017*; *Lee and Chen, 2004*; *Wiley et al., 1981*). The Spearman correlation between H1-specific HI titer (C) or H1-specific IFN-γ-secreting cell counts (D) with the H1 nucleotide divergence identified in individual pigs was calculated. The same statistics were also computed for H3 viruses between the H3-specific HI titer (G) or H3-specific IFN-γ-secreting cell counts (H) with the H3 nucleotide divergence identified in individual pigs. HA nucleotide divergence was calculated by summing the frequencies of all the HA SNVs identified in each sample. The HI titers of any individual pigs below the detection limit (1:10) are shown as 1:5. The number in specific dots represents the quantity of overlapping points plotted for pigs that had the same numbers for both variables, and the total quantity of available samples for this analysis is indicated as 'n.' All the within-host IAV nucleotide polymorphisms present in at least 1% of H1N1 and H3N2 sequencing reads from BALF samples are shown in *Figure 4—figure supplement 1*. The detailed information of identified H1N1 and H3N2 SNVs is displayed in *Figure 4—source data 1* and *Figure 4—source data 2*, respectively. The nonsynonymous SNVs that were identified at functional sites in pigs from different groups were summarized in *Figure 4—figure supplement 2*. The detailed results of permutation test on shared amino acid polymorphic sites in the 70 and 100% genome of H1N1 and H3N2 influenza A viruses (IAVs) can be found in *Figure 4—figure supplement 3*.

The online version of this article includes the following source data and figure supplement(s) for figure 4:

**Source data 1.** Summarized information of H1N1 single-nucleotide variants (SNVs) identified in vaccinated and unvaccinated pigs.

**Source data 2.** Summarized information of H3N2 single-nucleotide variants (SNVs) identified in vaccinated and unvaccinated pigs.

**Figure supplement 1.** Single-nucleotide variants (SNVs) of H1N1 and H3N2 viruses identified within pigs from different vaccination statuses.

**Figure supplement 2.** Detection of amino acid changes at functional sites.

**Figure supplement 3.** Shared amino acid polymorphisms sites in H1N1 and H3N2 influenza A viruses (IAVs).

and frequency of H1N1 SNVs by coding region and group are summarized in *Table 2*. The quantity and frequency of SNVs in each IAV coding region did not differ significantly among treatment groups (p=0.06–0.99 for SNVs quantity; p=0.06–0.80 for SNVs frequency, Kruskal-–Wallis rank-sum test). We found 75 out of 207 nonsynonymous mutations (mean frequencies with SD = 0.095 ± 0.221) located at functional relevant sites using the available annotations from the Sequence Feature Variant Types tool in the Influenza Research Database (*Noronha et al., 2012*; *Figure 4—figure supplement 2* and *Supplementary file 3*). The percentage and mean frequencies of these annotated SNVs were not significantly different between pigs from different treatment groups (percentage: p=0.101, chi-square test; frequency: p=0.473, Kruskal–Wallis rank-sum test). The H1N1 virus exhibited low antigenic variation regardless of treatment groups (*Figure 4B*). Only four different nonsynonymous (at nucleotide site 622 in Sb region found in three SINGLE LAIV and one NO VAC pigs, site 665 in Ca1 region from

**Table 2.** Average number and frequency of single-nucleotide variants (SNVs) detected on H1N1 sequences of bronchoalveolar lavage fluid (BALF) samples by coding region for each gene segment and treatment groups.

| Segment | PRIME BOOST (n = 12)* | | | SINGLE LAIV (n = 9) | | | NO VAC (n = 7) | | |
|---|---|---|---|---|---|---|---|---|---|
| | No. of sequences† | Mean no. of SNV (SD) | Mean SNV frequency (SD) | No. of sequences | Mean no. of SNV (SD) | Mean SNV frequency (SD) | No. of sequences | Mean no. of SNV (SD) | Mean SNV frequency (SD) |
| PB2 | 3 | 6.7 (4.0) | 0.117 (0.229) | 7 | 3.7 (2.6) | 0.162 (0.331) | 7 | 4.3 (4.9) | 0.135 (0.305) |
| PB1 | 2 | 4.5 (0.7) | 0.258 (0.393) | 7 | 2.7 (2.5) | 0.097 (0.228) | 7 | 5.3 (3.7) | 0.138 (0.292) |
| PB1-F2 | 2 | 0.5 (0.7) | 0.317 (NA) | 7 | 0.4 (0.5) | 0.249 (0.410) | 7 | 0.3 (0.8) | 0.018 (0.005) |
| PA | 4 | 2.5 (1.7) | 0.029 (0.024) | 7 | 3.0 (3.1) | 0.115 (0.274) | 7 | 2.9 (1.8) | 0.064 (0.100) |
| PA-X | 4 | 0.8 (1.0) | 0.035 (0.038) | 7 | 1.3 (1.1) | 0.221 (0.404) | 7 | 1.1 (1.3) | 0.058 (0.125) |
| HA | 3 | 4.7 (6.4) | 0.113 (0.259) | 7 | 3.0 (2.7) | 0.287 (0.390) | 7 | 3.6 (4.0) | 0.145 (0.323) |
| NP | 2 | 3.0 (1.4) | 0.236 (0.375) | 7 | 0.3 (0.5) | 0.014 (0.002) | 7 | 1.3 (1.4) | 0.035 (0.047) |
| NA | 1 | 5.0 (NA) | 0.036 (0.028) | 7 | 1.6 (1.1) | 0.059 (0.079) | 7 | 2.3 (2.7) | 0.079 (0.125) |
| M1 | 3 | 3.0 (1.0) | 0.029 (0.020) | 7 | 1.1 (1.3) | 0.249 (0.429) | 7 | 0.7 (1.1) | 0.256 (0.406) |
| M2 | 3 | 0.7 (0.6) | 0.017 (0.006) | 7 | 0.1 (0.4) | 0.196 (NA) | 7 | 0.1 (0.4) | 0.052 (NA) |
| NS1 | 3 | 0.7 (1.2) | 0.048 (0.001) | 7 | 1.1 (1.2) | 0.022 (0.017) | 7 | 0.9 (1.2) | 0.017 (0.011) |
| NS2 | 3 | 0.3 (0.6) | 0.049 (NA) | 7 | 0.9 (1.2) | 0.020 (0.008) | 7 | 0.6 (0.5) | 0.344 (0.407) |

SD, standard deviation; LAIV, live-attenuated influenza virus; VAC, vaccination; IAV, influenza A virus.

*'n' represents the number of BALF samples successfully sequenced by Illumina in each of the treatment groups.

†The 'No. of sequences' columns represent the number of sequences from H1N1 IAVs for any given gene product from the total available samples (n) within each treatment group.

a PRIME BOOST pig, site 473 in Ca2 region from a NO VAC pig, and site 263 in Cb region from a NO VAC pig) nucleotide changes were identified in H1 antigenic regions in one PRIME BOOST, three SINGLE LAIV, and two NO VAC pigs. The divergence of H1 genes did not differ based on the intensity of the humoral and cellular immunity induced by vaccination (*Figure 4C and D*). There was no evidence of correlation of the H1 nucleotide divergence with the H1-specific HI titers ($R = 0.069$, p=0.79, Spearman's rank-order correlation test) nor with the H1-specific IFN-γ-secreting cell spots ($R = –0.153$, p=0.62, Spearman's rank-order correlation test).

Within the SNVs dataset of the H3N2 virus, we observed 364 SNVs out of 257,699 sequenced consensus nucleotides (205 nonsynonymous, 153 synonymous, 6 stop-gained) in 21 BALF samples (*Figure 4—figure supplement 1B*, *Figure 4—source data 2*). Similar to H1N1, the H3N2 SNVs were dominated by low-frequency variants (*Figure 4E*). There were 187 (91.2%) nonsynonymous, 120 (78.4%) synonymous, and 6 (100.0%) stop-gained SNVs whose frequency was below 10%. About 4.4% (16/364) of H3N2 SNVs were presented at the consensus level, which was less than the SNVs presented for the H1N1 virus (p=0.007, chi-square test). We summarized the SNVs quantity and frequency of the H3N2 virus by coding region and group in *Table 3*, and no statistical differences were detected in the average SNV number between treatment groups (p=0.27–1.00, Kruskal–Wallis rank-sum test). The differences of SNVs frequencies in each gene segment were not significant between treatment groups, except the frequency of N2 SNVs detected in PRIME BOOST pigs was lower than that of SINGLE LAIV pigs (p=0.046, Dunn's test with Benjamini–Hochberg correction). There were 93 nonsynonymous SNVs (mean frequencies with SD = 0.048 ± 0.117) located at functional relevant sites in the H3N2 genomes. We found that the percentage and average frequencies of these functional annotated SNVs were similar between treatment groups (percentage: p=0.122, chi-square test; frequency: p=0.058, Kruskal–Wallis rank-sum test) (*Figure 4—figure supplement 2* and *Supplementary file 4*). Consistent with the H1N1 virus, we did not find a significant impact of immunity on H3 variants (*Figure 4F–H*). Specifically, there were only two nonsynonymous (at nucleotide site 830 in E region from a NO VAC pig and site 961 in C region from a PRIME BOOST pig) nucleotide changes located in the H3 antigenic regions in one PRIME BOOST and one NO VAC pigs, and we did not detect any associations between the H3 nucleotide divergence and H3-specific HI titers ($R = –0.203$, p=0.39, Spearman's rank-order

**Table 3.** Average number and frequency of single-nucleotide variants (SNVs) detected on H3N2 sequences of bronchoalveolar lavage fluid (BALF) samples by coding regions and treatment groups.

| Segment | PRIME BOOST (n = 12)* | | | SINGLE LAIV (n = 9) | | | NO VAC (n = 7) | | |
|---|---|---|---|---|---|---|---|---|---|
| | No. of sequences† | Mean no. of SNV (SD) | Mean SNV frequency (SD) | No. of sequences | Mean no. of SNV (SD) | Mean SNV frequency (SD) | No. of sequences | Mean no. of SNV (SD) | Mean SNV frequency (SD) |
| PB2 | 10 | 4.9 (3.8) | 0.155 (0.287) | 6 | 4.0 (2.9) | 0.130 (0.259) | 4 | 3.3 (0.5) | 0.147 (0.162) |
| PB1 | 9 | 3.1 (2.7) | 0.119 (0.231) | 6 | 3.2 (3.8) | 0.044 (0.038) | 3 | 2.3 (1.2) | 0.024 (0.019) |
| PB1-F2 | 9 | 0.1 (0.3) | 0.024 (NA) | 6 | 0.0 (0.0) | 0.000 (0.000) | 3 | 0.0 (0.0) | 0.000 (0.000) |
| PA | 8 | 2.5 (2.7) | 0.060 (0.096) | 6 | 2.2 (1.8) | 0.054 (0.106) | 4 | 2.0 (0.8) | 0.033 (0.028) |
| PA-X | 8 | 0.8 (0.9) | 0.063 (0.069) | 6 | 0.7 (1.2) | 0.015 (0.003) | 4 | 0.5 (0.6) | 0.038 (0.011) |
| HA | 10 | 1.2 (1.2) | 0.036 (0.060) | 6 | 1.7 (0.8) | 0.092 (0.179) | 4 | 1.0 (0.0) | 0.024 (0.015) |
| NP | 11 | 3.0 (3.1) | 0.102 (0.246) | 6 | 1.7 (1.2) | 0.080 (0.169) | 4 | 1.0 (0.8) | 0.023 (0.008) |
| NA | 11 | 2.3 (1.9) | 0.019 (0.012) | 6 | 1.8 (1.7) | 0.035 (0.023) | 4 | 1.8 (1.5) | 0.170 (0.256) |
| M1 | 11 | 1.3 (1.7) | 0.041 (0.079) | 6 | 0.7 (1.2) | 0.017 (0.011) | 4 | 0.0 (0.0) | 0.000 (0.000) |
| M2 | 11 | 0.5 (0.8) | 0.032 (0.024) | 6 | 0.3 (0.5) | 0.018 (0.004) | 4 | 0.0 (0.0) | 0.000 (0.000) |
| NS1 | 10 | 1.3 (1.4) | 0.078 (0.110) | 6 | 0.3 (0.5) | 0.033 (0.016) | 4 | 1.0 (0.8) | 0.014 (0.003) |
| NS2 | 10 | 0.5 (0.5) | 0.122 (0.167) | 6 | 0.3 (0.8) | 0.052 (0.059) | 4 | 0.8 (1.0) | 0.036 (0.032) |

SD, standard deviation; NA, not applicable; LAIV: live-attenuated influenza virus; IAV, influenza A virus.

*'n' represents the number of BALF samples successfully sequenced by Illumina by treatment groups.

†The 'No. of sequences' columns represent the number of sequences from H3N2 IAVs for any given protein from the total available samples (n) within each treatment group.

correlation test) or H3-specific IFN-γ-secreting cell counts (R = –0.046, p=0.87, Spearman's rank-order correlation test). The detailed results of H1/H3 HI titers and H1/H3-specific IFN-γ-secreting cell counts for each vaccinated and nonvaccinated pig can be found in *Li et al., 2020*.

Vaccine-induced immunity may drive genetic selection within a specific genetic region or site of the IAV genome, facilitating the same amino acid changes in multiple pigs even if the pigs are housed in different rooms, which would suggest a sign of convergent evolution. There were 0, 1 (PB1 V632A), and 2 (HA A242G and PB2 N562I) amino acid changes of H1N1 viruses in at least two PRIME BOOST, SINGLE LAIV, and NO VAC pigs from different rooms, respectively. For the H3N2 virus, we only identified NA G143R in NO VAC pigs from multiple rooms. We performed the permutation test for each treatment group to evaluate whether the number of shared amino acid site changes observed in pigs from multiple rooms was more than expected by chance (*Figure 4—figure supplement 3*). However, we found that the shared polymorphisms in pigs of all three treatment groups were no more than expected by chance for both H1N1 and H3N2 viruses, except for the weak genome convergence of H3N2 viruses observed in NO VAC pigs (permutation in 100% IAV genome: p=0.03536, 70% IAV genome: p=0.04907). As the permutation test was based on the assumption that all the four shared amino acid changes identified in multiple pigs from different rooms were developed independently, the presence of IAV shared polymorphisms in pigs may even be more scarce if any of them was the minor variant transmitted from the seeders. Overall, we observed limited shared diversity in H1N1 and H3N2 IAV detected in vaccinated pigs, which shows a minimal effect of vaccination on the degree of IAV convergence in pigs from the different rooms for either the H1N1 or the H3N2 virus.

## Within-pig nucleotide polymorphisms and evolutionary rates of H1N1 and H3N2 viral populations are similar regardless of vaccination status

We calculated the nucleotide diversity (Pi, average number of pairwise nucleotide differences per site) to measure the degree of genetic variation for the H1N1 and H3N2 viral populations within pigs from the different treatment groups (*Figure 5A*). However, neither the H1N1 nor the H3N2 viruses from any pig, regardless of treatment group, had statistically significant Pi differences at individual coding regions or in all coding regions combined (i.e., whole genome). One exception was the higher nucleotide diversity in pigs from the PRIME BOOST group when compared to the SINGLE LAIV group for the NP gene (p=0.049, Dunn's test with Benjamini–Hochberg correction) of the H1N1 virus.

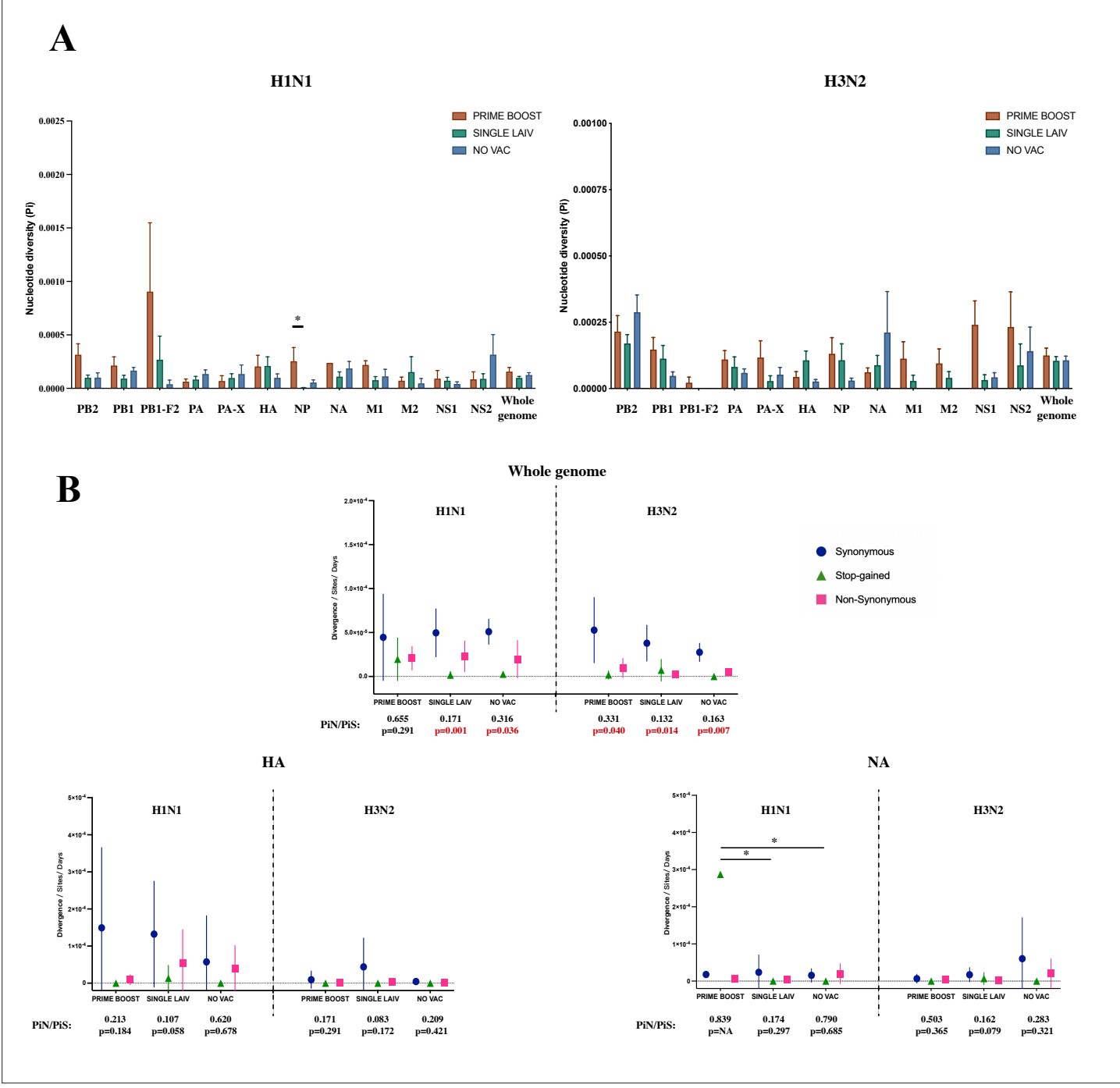

**Figure 5.** Within-host nucleotide diversity and evolutionary rate of H1N1 and H3N2 influenza A virus (IAV) by coding regions. (**A**) Nucleotide diversity (Pi) was computed for each coding region of H1N1 and H3N2 viruses for each treatment group. The nucleotide diversity is shown as mean with standard error for PRIME BOOST (brown), SINGLE LAIV (dark green), and NO VAC (dark blue) pigs. The statistical results are noted (*p<0.05) if the nucleotide diversity of any coding region differed significantly between treatment groups, which were compared by Kruskal–Wallis rank-sum test; the Dunn's test was used for the multiple pairwise comparisons with Benjamini–Hochberg correction. The standard errors were calculated through 10,000 bootstrap resampling with the replacement. (**B**) The evolutionary rates were calculated separately for H1N1 and H3N2 viruses at synonymous (dark blue circle), nonsynonymous (pink square), and stop-gained (green triangle) sites for each sample on antigenic proteins and whole-genome level. The evolutionary rates are displayed as means with standard deviations and compared by Kruskal–Wallis rank-sum test, followed with the Dunn's test for the multiple pairwise comparisons; the p-values were corrected by the Benjamini–Hochberg method. The statistical results are noted (*p<0.05) if the evolutionary rate of any coding region and mutational type was significantly different between pigs from any two groups. The nonsynonymous (piN) or synonymous (PiS) nucleotide diversity is the average number of pairwise nonsynonymous or synonymous polymorphism/diversity per nonsynonymous or synonymous

*Figure 5 continued on next page*

*Figure 5 continued*

site and computed for each coding region and sample by treatment groups. The paired *t*-test was used to test the null hypothesis that piN = piS, and the significant results (p<0.05) are marked in red font. The p-values are assigned as 'NA' if no SNVs were identified in a given coding region in any groups or the number of available samples was not enough to perform the statistical analysis, and the ratios of piN to piS (piN/piS) are displayed as 'NA' if there was no synonymous single-nucleotide variant (SNV) found in that coding region for pigs in any treatment groups. The piN/piS = 0 indicates no nonsynonymous SNVs were identified in any coding region from pigs in any treatment groups. The number of samples/sequences used to calculated the nucleotide diversity and evolutionary rates for each H1N1 and H3N2 coding region from different groups can be found in *Table 2* and *Table 3*, respectively. *Figure 5—figure supplement 1* shows the evolutionary rates and ratios of nonsynonymous to synonymous nucleotide diversity on protein products from H1N1 and H3N2 internal genes. The detailed values of PiN and PiS for H1N1 and H3N2 viruses by treatment groups and coding regions can be found in *Figure 5—source data 1*.

The online version of this article includes the following source data and figure supplement(s) for figure 5:

**Source data 1.** Values of nonsynonymous (PiN) and synonymous (PiS) nucleotide diversity in H1N1 and H3N2 challenge viruses by coding regions and treatment groups.

**Figure supplement 1.** The evolutionary rate and ratio of nonsynonymous to synonymous nucleotide diversity (PiN/PiS) of H1N1 and H3N2 influenza A virus (IAV) on coding regions located in internal genes.

We calculated the evolutionary rates for H1N1 and H3N2 viruses at the whole-genome and individual coding region level to assess the pace of synonymous, nonsynonymous, and stop-gained mutations accumulated on virus populations in pigs by treatment (*Figure 5B*, *Figure 5—figure supplement 1*). Only the rate of stop-gained mutations in the N1 gene differed between groups (PRIME BOOST – SINGLE LAIV: p=0.0009, PRIME BOOST – NO VAC: p=0.0004, Dunn's test with Benjamini–Hochberg correction). For all other genome changes in the H1N1 and H3N2, similar rates for all three types of mutations between any two individual groups did not differ significantly in any individual IAV coding region within pigs from different vaccination statuses.

The comparison between values of nonsynonymous nucleotide diversity (piN) and synonymous nucleotide diversity (piS) is a measure of within-host virus diversity that can be used to infer the types of selective forces from positive (Darwinian) selection (piN/piS > 1), purifying (negative) selection (piN/piS < 1), and genetic drift (piN/piS ~ 1). The detailed values of piN and piS for each treatment group and coding region are summarized in *Figure 5—source data 1*. At the whole-genome level, both H1N1 and H3N2 viruses exhibited significant piN < piS within pigs from all three groups, except for H1N1 in PRIME BOOST pigs (piN/piS = 0.655, p=0.291, paired *t*-test), which showed signs of purifying selection (*Figure 5B*). For the majority of IAV coding regions, IAV nucleotide diversity at the individual coding region level exhibited piN < piS without showing significant differences for the H1N1 or the H3N2 viruses within pigs regardless of treatment groups. The exceptions for IAV nucleotide diversity at the coding region level were found for these few within-host instances, for example, H1N1 PB2 gene in SINGLE LAIV (piN/piS = 0.066, p=0.021, paired *t*-test) and NO VAC (piN/piS = 0.291, p=0.047, paired *t*-test) pigs, the H3N2 PB2 gene in PRIME BOOST (piN/piS = 0.198, p=0.036, paired *t*-test) and SINGLE LAIV (piN/piS = 0.082, p=0.014, paired *t*-test) pigs, and H1N1 PB1 gene in NO VAC (piN/piS = 0.223, p=0.036, paired *t*-test) pigs (*Figure 5—figure supplement 1*). Moreover, we did not find any evidence of positive selection in any coding region for both H1N1 and H3N2 viruses in any of the pigs.

When computing the correlations between the intensity of swine humoral (HA-specific HI titers) and cellular (HA-specific IFN-γ-secreting cell counts) immunity versus the IAV HA Pi, PiN, or PiS values of individual pigs for both H1N1 and H3N2 viruses, only a weak negative association was found between the H3-specific IFN-γ ELISPOT cell counts and the values of PiN for H3 genes (R = –0.512, p=0.043, Spearman's rank-order correlation test). For H1 cellular and humoral immunity and H3 humoral immunity, no correlations were detected (*Supplementary file 5*).

## Discussion

Understanding the impact of vaccination on IAV within-host evolution is critical for improving overall animal health and productivity while also decreasing the emergence of novel antigenic variants with pandemic potential (*Diaz et al., 2017*; *Nirmala et al., 2021*; *USDA, 2016*). To better mimic field scenarios, we set up an in vivo co-infection model where we challenged naïve pigs with either an H1N1 (gamma clade 1A.3.3.3) or an H3N2 (human-like clade 3.2010.1) virus belonging to the

dominant subtypes and lineages identified during US pig surveillance (*Anderson et al., 2013*). Pigs were co-housed in contact with nonvaccinated pigs or pigs vaccinated by licensed multivalent vaccines following multiple vaccination protocols (*Li et al., 2020*). The co-infection model allowed us to simultaneously assess the IAV genetic variation and reassortment under immune pressure in pigs and provided a rare view of the full impact of vaccination on IAV evolution. The dataset includes pigs with different levels of virus shedding and immune responses that help us define factors that affect IAV evolution (*Li et al., 2020*). We directly sequenced and performed the isolation of viral plaques on selected BALF samples without additional virus propagation in cells to ensure that no mutation or reassortants were developed during cell culture. Our results suggested that the pigs receiving two doses of vaccines (PRIME BOOST) had a lower proportion of IAV reassortants in fewer days when the pigs were co-infected with both challenge viruses compared to nonvaccinated pigs. A significant number of mutations with abundant functional relevant amino acid changes were present within pigs in H1N1 and H3N2 viral populations regardless of vaccination status. However, we did not detect a major effect of vaccination on IAV within-host diversity as there were few antigenic variants detected in the IAV populations, and we found no differences in the number and frequency of identified SNVs, the evolutionary rates, nor in the nucleotide diversity of IAV in pigs of the different treatment groups.

Within the 7 days of the study period, we identified a large number of IAV reassortants with multiple distinct genotypes in the pigs that had been co-infected with the H1 and H3 viruses. This finding is particularly notable when considering the complex ecosystem and management practices implemented in swine farms, and the pigs with various vaccination and infection statuses present in the farms (*Garrido-Mantilla et al., 2021*; *Reynolds et al., 2014*; *White et al., 2017*). The fluidity of ages and immune statuses of the pigs in swine farms enable viruses to continually infect susceptible pigs and create an ideal breeding ground for IAV to reassort and circulate (*Diaz et al., 2017*; *Chamba Pardo et al., 2018*). Given the background that the coexistence of viruses with distinct lineages and genotypes is common in pig populations, the frequency of IAV reassortment could be extensive, especially considering that most of the pigs need to be housed approximately for 24–26 weeks in farms (*Nirmala et al., 2021*). Our study provides initial evidence that vaccination can play a role in decreasing the generation and emergence of new reassortant viruses in pigs. These results, if confirmed in field conditions, show the significance of IAV vaccination in pig populations not only by restricting the virus circulation and clinical impact but also by reducing the IAV genotypic diversity that is important for the overall IAV control at the human–animal interface (*Dwyer and Kirkland, 2011*).

The factors driving reassortment are complicated, and for IAV reassortment to occur two parental viruses need to reach multiple criteria (*Lowen, 2018*). The co-infection of distinct IAV viruses at the cellular level is an essential prerequisite for IAV reassortment (*Marshall et al., 2013*). Numerous factors affecting IAV cellular co-infections have been elucidated in vitro or under experimental conditions, including the IAV challenge dose, the time interval between primary and secondary infection, and the location of initial virus replication (*Richard et al., 2018*; *Tao et al., 2014*). Our study found that pigs co-infected for a longer duration tended to generate more reassortants, which showed that vaccination could minimize IAV reassortment by reducing the possibility of pigs being co-infected by distinct viruses. It has been shown that most IAV reassortants emerge with lower fitness compared with parental viruses, and evidence suggests that the strength of negative selection on the reassortants is positively associated with the genetic distance between their parental strains (*Villa and Lässig, 2017*). Tissue tropism also appears to play a role in the generation of novel reassortants in pigs, with most reassortants generated in pigs being recovered in the middle and lower respiratory tissues. In contrast, fewer reassortants were identified in the nasal cavity (*Zhang et al., 2018*). It is likely that the high temperature and presence of various types of sialic acid receptors in the lungs benefit infection and replication of the reassortant viruses (*de Graaf and Fouchier, 2014*; *Nelli et al., 2010*; *Zhang et al., 2018*), which might help explain the significant number of IAV reassortants we identified in the BALF samples of this study. We observed that the IAV internal genes were exchanged fairly easily between the two challenge, that is, parental, viruses compared to the antigenic genes. Considering the nucleotide homology of internal genes between the challenge viruses was much higher (over 90%) than that of the antigenic genes (~53%), the mismatch of package signals and functional proteins could explain these results given that these restraining factors facilitate virus reassortment between similar strains rather than in distantly related viruses (*Lowen, 2017*; *Lowen, 2018*; *Richard et al., 2018*). Besides, there are

other environmental conditions and other host factors that may affect the fitness of the reassortants in a populations (*Lowen, 2018*).

The high plasticity of the IAV genome and the fast-expanding nature of IAV favors the generation of progeny virions with slightly different mutation 'signatures' (*Lauring and Andino, 2010*; *Martínez et al., 2012*). Although most progeny variants have a deleterious effect on the overall fitness of the viral populations, some of the beneficial mutants may go through positive selection and can quickly dominate after sudden changes in the environment thereby shifting the fitness landscape, which may maximize the possibility of IAV to replicate in harsh conditions (*Martínez et al., 2012*). As a result, IAV may increase mutation rates at the cost of fitness to enrich the genetic pool and accumulate more beneficial variants with high frequencies favored by positive selection (*Elena and Sanjuán, 2005*; *Sprouffske et al., 2018*). In a stable environment, gaining fitness is a priority for IAV rather than the accumulation of genetic variation at the consensus level (*Martínez et al., 2012*). Therefore, purifying selection takes place as the major selective force that continually removes new variants and generates IAV populations with a high proportion of low-frequency variants (*Moncla et al., 2020*). Previous studies have identified that host immunity could drive positive selection on the IAV antigenic proteins in the long term (*Bush et al., 1999*; *Li et al., 2011*; *Nelson and Holmes, 2007*). However, in our study, we rarely observed the inference of the same selection pressures at any coding regions for the H1N1 and H3N2 viruses, which is consistent with the results of within-host diversity on H5N1 virus (*Moncla et al., 2020*). Instead, the within-host diversity of both subtypes was dominated by low-frequency variants and exhibited similar or higher synonymous polymorphisms compared to the nonsynonymous polymorphisms. The variation patterns of IAV suggested that the viral populations may be broadly shaped by purifying selection and genetic drift within pigs regardless of vaccination statuses. In addition to the close mutational spectra we observed on H1N1 and H3N2 viruses in pigs from various treatment groups, we detected limited shared diversity, which developed independently in at least two PRIME BOOST, SINGLE LAIV, or NO VAC pigs for both viruses. Our observation suggested that vaccination did not drive the progress of IAV genome convergence, at least in the swine lower respiratory tract, when we measured the virus's shared diversity between pigs, which in turn reflected the limited selective forces raised by vaccination on viral within-host diversity.

The HA protein is the major target of the host adaptive immune response and crucial to IAV evolution. Within our dataset, there were few amino acid changes that fell in the H1 and H3 antigenic regions. Furthermore, the nucleotide divergence of the HA segments did not correlate with the intensity of humoral and cellular immune responses induced by vaccination. These observations are concordant with the results from *Debbink et al., 2017*, which showed the limited effect of vaccination on antigenic diversity of H3N2 viruses in humans. However, other factors like host type, IAV strain, infection dose, challenge/infection model, virus passage among animals, and observation period may cause discrepancies in the results of selection of the antigenic variants between the studies that explore the influence of immune pressure on IAV within-host diversity (*Debbink et al., 2017*; *Hoelzer et al., 2010*; *Murcia et al., 2013*; *Murcia et al., 2012*). Taken together, we did not find evidence that vaccination influenced IAV genetic variation in the lower respiratory tract of pigs. The observation of our study aligns with the data from *Murcia et al., 2012*, which suggested that vaccine-induced immunity had a minimal impact on the genetic variation of IAV populations in both upper and lower respiratory tracts of pigs.

Evaluating IAV evolution in vaccinated hosts is often difficult to perform. One challenge is to obtain enough qualifying samples in pigs that are well-protected by vaccination as there may be low levels of, or even no, replicating virus. The lack of virus detection in samples from vaccinated pigs may introduce sample selection bias. That is, samples from vaccinated pigs with replicating virus may represent pigs with poor immunity that may not adequately reflect the effect of vaccination on IAV within-host evolution. However, in our study this may not be the case since samples submitted for direct sequencing or plaque assay from PRIME BOOST pigs had significantly higher H1 and H3-specific antibody titers compared to SINGLE LAIV or NO VAC pigs. Besides, our findings on IAV within-host diversity in vaccinated and unvaccinated pigs agree with those of similar studies in pigs and other hosts (*Debbink et al., 2017*; *Murcia et al., 2013*; *Murcia et al., 2012*). Our results may have also been influenced by the degree of shedding of the seeder pigs in each of the rooms as some pigs may have shed more than others, which could influence the likelihood of co-infections to occur in the in-contact pigs, biasing the likelihood of detecting reassortants. The viruses from the seeders may also contain

subgenomic variants that transmit to in-contact pigs. We found that the three super-reassortant pigs (pig #4479, #4490, and #5174) came from three different rooms, which shows that IAV reassortment did not have a 'room effect.' In addition, it was not the study's primary aim to detect the selection of individual IAV SNVs or reassortants specific to any vaccination protocol but rather to compare the impact of vaccination on the overall evolutionary trends of virus populations in pigs. The PRIME BOOST samples analyzed in this study included samples from pigs receiving multiple prime-boost vaccination protocols. Different vaccine combinations may introduce variability in the profiles of antibody populations raised among PRIME BOOST pigs. In our study, this did not seem to cause significant differences between groups because pigs in different groups had similar HA-specific adaptive immune responses and exhibited a comparable level of protective efficacy on pigs against the challenge viruses (*Li et al., 2020*). However, whether and how the different homology between vaccine and challenge viruses, especially in prime-boost vaccination schemes, will change the IAV evolutionary process within pigs still needs further exploration. We also had a limited number of BALF samples available for plaque purification and the number of plaques isolated from each sample was not equal. Although we acknowledge this as a limitation, the overall conclusions were supported by the statistical methods employed to assess the relationship between the percent of reassortants and the treatment group at the individual pig level, which accounted for the impact of unequal sample quantity from different treatment groups and the variability of total plaques isolated from each sample. As our statistical analysis relies on the asymptotic theory, which assesses how the estimators or tests behave as the sample size becomes larger (*Höpfner, 2014*), further research with a larger number of samples collected from both experimental and field settings should help validate the concept of using IAV vaccines to mitigate IAV reassortment in pigs.

Finally, understanding the association between IAV vaccination and evolution is hard but necessary. IAV within-host evolution plays a vital role in disease outcomes since mutations and reassortment events can significantly influence the biological characteristics of IAV. In this study, we conclude that pig vaccination should be explored as a way to reduce the generation and emergence of reassortant viruses, and this may be particularly important in farm animals where large populations are housed together. In contrast, we did not observe the same significant effect of vaccination on IAV genetic variation. Our research results provide insights into the complexity of IAV evolution in pigs and will help develop more effective influenza control programs to mitigate the burden of IAV infections in pigs while decreasing the risk of zoonotic infections and preserving public health. We believe our study assembles a comprehensive recognition of IAV evolutionary strategies under immune pressure but does not fully reflect the situation in the field. Therefore, more studies are needed to verify the extent of IAV mutations and reassortment in vaccinated and unvaccinated pigs and understand what factors contribute to the transmission of reassortant viruses under field conditions, which is essential for developing IAV vaccines and surveillance strategies.

## Materials and methods
### Vaccine-challenge experiments in pigs
All the BALF samples analyzed in this study were obtained from pigs infected using a co-infection seeder pig model using an H1N1 and an H3N2 IAV strain to evaluate the protective efficacy of different prime-boost vaccination protocols in pigs (*Li et al., 2020*). All the animal work followed the protocols approved by the University of Minnesota Institutional Animal Care and Use Committee (protocol ID: 1712-35407A) and the Institutional Biosafety Committee (protocol ID: 1508-32918H).

The infection seeder pig model was set up to mimic field conditions where pigs become infected by contact transmission by challenging 14 eight-week-old unvaccinated pigs (seeder pigs) with either H1N1 (A/swine/Minnesota/PAH-618/2011) or H3N2 (A/swine/Minnesota/080470/2015) IAV strains. After challenge, seeder pigs were kept separated by subtype until they were confirmed to be shedding IAV at 2 days post inoculation (DPI). At that time, one seeder of each subtype was placed in seven rooms to serve as the infection source of H1 and H3 viruses to 10 other pigs from different treatment groups to attempt simultaneous in-contact exposure to both strains and subsequent co-infection. The homology of nucleotides and amino acids between the genes and protein products of H1N1 and H3N2 challenge viruses is summarized in *Supplementary file 6*. Whole-genome sequences of the two challenge viruses have been deposited in GenBank with accession numbers from MT377710 to

MT377725. These deposited sequences were obtained from the nasal swabs taken from the seeder pigs (pig #5168 H1N1 seeder and pig #4947 H3N2 seeder) at 2 DPI before commingling with the rest of the pigs and used as reference sequences for variant identification. We assumed these consensus sequences are a good representation of the genomes of the two challenge viruses at the time transmission to contact pigs started. The vaccines used for different vaccination protocols included an inactivated commercial quadrivalent vaccine (COM), an inactivated autogenous trivalent vaccine (AUT), and a live-attenuated bivalent vaccine (LAIV). The AUT contains the vaccine strains with the highest HA amino acid identity with both challenge viruses (H1N1 vaccine – H1N1 challenge: 96.5%, H1N2 vaccine – H1N1 challenge: 78.6%, H3N2 vaccine – H3N2 challenge: 99.1%). In contrast, the LAIV strains had low HA amino acid identity with the challenge strains (H1N1 vaccine – H1N1 challenge: 93.3%, H3N2 vaccine – H3N2 challenge: 89.2%). While one of the H1 COM vaccine strains had high HA amino acid homology with the H1N1 challenge virus (H1N1 vaccine – H1N1 challenge: 95.1%, H1N2 vaccine – H1N1 challenge: 78.4%), both of its two H3N2 vaccine strains had low HA homology compared to the H3N2 challenge virus (H3N2 vaccine 1 – H3N2 challenge: 87.1%, H3N2 vaccine 2 – H3N2 challenge: 88.2%). The detailed information for the vaccines used in this study can be found in *Li et al., 2020*. Seventy pigs were randomly assigned to seven treatment groups with different vaccine combinations that included four whole inactivated vaccinated (WIV) groups (COM/COM, AUT/AUT, AUT/COM, and COM/AUT), two live-attenuated vaccinated (LAIV) groups (LAIV/NONE and LAIV/COM) and one positive unvaccinated control group (NO VAC/CHA). No samples from AUT/COM pigs were used in the current study as pigs were protected by vaccination and no IAV-positive samples were obtained. Pigs were primed with the first vaccine administration at approximately 3 weeks of age, boosted at 6 weeks of age, commingled with seeder pigs at 8 weeks of age, and euthanized at 9 weeks of age. Pigs vaccinated with inactivated vaccines (COM or AUT) and the positive control groups were housed together in the same rooms and evenly distributed in five rooms (two pigs/ treatment/room). Pigs receiving the LAIV treatment were evenly distributed in two rooms (five pigs/ treatment/room). Each room contained 2 seeders and 10 treatment pigs for a total of 12 pigs per room. Nasal swabs were collected daily for all the pigs from 2 to 6 DPC. BALF samples were taken at necropsy at 7 DPC with the seeder pigs. HA subtyping real-time PCR was performed in all the nasal swabs and BALF samples using the VetMAX gold SIV subtyping kit (Life Technologies, Austin, TX). The methods and results for testing the BALF samples by IAV matrix gene real-time RT-PCR, virus titration, IFN-γ ELISPOT cell counts on the lymph nodes and hemagglutinin inhibition (HI) titers for the serum obtained from each pig are shown in *Li et al., 2020*. The BALF samples with Ct values below 35 obtained using the IAV matrix gene real-time RT-PCR were selected for whole-genome sequencing directly from the sample, without any virus propagation. Overall, we obtained the IAV genomes from 28 BALF samples to identify the IAV within-host diversity in pigs (*Table 1*).

To evaluate IAV reassortment happening in naïve and vaccinated pigs, we selected pigs from two rooms housing pigs vaccinated with inactivated vaccines and one room housing LAIV pigs. This was done to select seeder pigs that shed relatively high quantity of both challenge viruses (Ct values ranging between 21.15 and 30.51 when placed in contact with the vaccinated pigs) and that shed IAV for at least 2 days after being mixed with the other pigs. This situation allowed for sufficient IAV contact exposure from the seeders to all pigs in the room, which is the prerequisite for IAV reassortment. BALF samples from pigs that yielded virus replication by virus titration (TCID50 ≥ 1.75 / ml) were used for viral plaque purification. Thirteen BALF samples yield IAV plaques that then were used to evaluate IAV reassortment happening in the naïve and vaccinated pigs (*Table 1*). For analysis purposes, pigs were grouped in three groups named (a) prime boost (COM/COM, AUT/AUT, COM/ AUT and LAIV/COM pigs), (b) single LAIV (LAIV/NONE pigs), and (c) nonvaccinated (NO VAC/CHA pigs) groups.

## Plaque library preparation

Plaque assay to purify individual virions was performed on Madin–Darby canine kidney (MDCK) cell monolayers (*Tobita et al., 1975*). MDCK cells were obtained from the University of Minnesota Veterinary Diagnostic Laboratory (VDL). The cells were purchased from the American Type Culture Collection (ATCC-CCL-34). The cell line identification was verified morphologically. The MDCK cells were free of mycoplasma contamination and tested annually by VDL staff. Briefly, BALF samples were tenfold serially diluted using IAV growth medium, which was made up of Dulbecco's modified

Eagle medium (DMEM, Gibco Invitrogen, Carlsbad, CA), 4% BSA fraction V 7.5% solution (Gibco, Life Technologies, Carlsbad, CA), 0.15% 1 mg/ml TPCK trypsin (Sigma-Aldrich, St. Louis, USA), and 1% antibiotic-antimycotic (Gibco, Life Technologies, New York, NY). The MDCK cells cultured in six-well plates, washed twice using Hanks' Balanced Salt Solution (HBSS, BioWhittaker, Verviers, Belgium) with 0.15% 1 mg/ml TPCK trypsin (Sigma-Aldrich), then inoculated using the diluted BALF samples and incubated for 1 hr in 37°C at 5% $CO_2$ incubator. After melting in 70°C water bath, one volume of 4% Agarose Gel (Gibco, Life Technologies) was mixed by a three-volume preheated IAV growth media to make the overlay gel liquid and kept at 37°C water bath. After 1 hr incubation, the diluted samples were aspirated and gently replaced by the warm overlay gel liquid at room temperature. When the overlayed gel became solid, the plates were invertedly incubated in a $CO_2$ incubator for 3–5 days. Up to 30 visualized plaques were randomly picked up from each sample using micropipette tips and propagated individually on MDCK cells. The isolated plaques were stored at –80°C for further whole-genome sequencing. The number of plaques isolated from each BALF sample is summarized in *Supplementary file 2*.

## RNA extraction, next-generation sequencing, and quality control

RNA extraction from BALF samples and plaque isolates was performed using the MagMax Viral RNA isolation kit (Ambion, Life Technologies, USA) (*Zhou et al., 2009*). The viral cDNA was amplified from extracted RNA through one-step reverse transcription-PCR amplification by using SuperScript III One-Step RT-PCR system with High Fidelity Platinum Taq DNA Polymerase (Invitrogen, Life Technologies, USA) with degenerate primers (10 uM MBTuni-12M and MBTuni-13) (*Keller et al., 2018*; *Zhou et al., 2009*). The PCR product was checked by NanoDrop 1000 (Thermo Fisher Scientific) and cleaned up by the QIAGEN QIAquick PCR Purification Kit (QIAGEN, USA). Purified PCR products were fragmented and tagged with the indexed adaptors using the Nextera DNA XT Sample Preparation Kit (Illumina, San Diego, CA). The sequence library was quantified by using the Quant-iT PicoGreen dsDNA Assay Kit (Invitrogen). The barcoded library was pooled in equimolar concentrations and multiplexed and sequenced by 2 × 150 bp paired-ends on Illumina Nextseq 550 Mid-Output mode platform at the University of Minnesota Genomics Center (UMGC). The sequence data obtained from UMGC was analyzed using the software available at the University of Minnesota Supercomputing Institute (MSI) platform. The raw reads were assessed for sequence quality using Fast-QC and trimmed by Trimmomatic (*Andrew, 2010*; *Bolger et al., 2014*). The trimming on the BALF sequences was conducted using the command: java -jar path-to-trimmomatic-0.33.jar PE input.fastq output.fastq ILLUMINA-CLIP: path-to-adaptor.fasta:2:30:10 LEADING:3 TRAILING:3 SLIDINGWINDOW:5:30 MINLEN:75. The command removes the adaptors and the bases with Q-score below 3 at the beginning and end of the raw reads, trimming the sequences in sliding windows of 5 base pairs and cutting the scan bases with the average Q-scores under 30. The reads were discarded if trimmed to the length below 75 bps. The trimming of the plaque sequences was performed using the same command but with different parameters: java -jar path-to-trimmomatic-0.33.jar PE input.fastq output.fastq ILLUMINACLIP: path-to-adaptor.fasta:2:30:10 LEADING:3 TRAILING:3 SLIDINGWINDOW:4:20 MINLEN:36, which trimmed the raw reads with the sliding windows of 4 base pairs and cutting the scan bases with the average Q-scores under 20. Reads were omitted if the length was below 36 after trimming.

## Reassortant recognition of viral plaques

The trimmed reads of viral plaque isolates were de novo assembled by Shovill (*Seemann, 2017*). The assembled contigs were annotated by OctoFLU for the initial inspection and sorted by IAV genes (*Chang et al., 2019*). Within each sample, the same IAV gene's contigs were omitted if the sequence length was <70% of the total nucleotide bases or <20% of the overall k-mer coverage on corresponding IAV segments. Occasionally, the longest contigs (~1.7% of total analyzed sequences) of any plaque isolates were preserved for genotyping if no sequences covered 70% of nucleotide sites for any of their IAV segments. The consensus sequences from isolated plaques for IAV reassortant identifications were shown in *Figure 2—source data 2*, and aligned with the sequences from the curated reference package from OctoFLU, and the sequences of H1N1 and H3N2 challenge viruses were aligned using the MUSCLE program in Geneious (version 2021.0.3) (*Chang et al., 2019*; *Edgar, 2004*; *Kearse et al., 2012*). The plaque gene segments' sources were initially checked by comparing the genetic distance between plaque sequences with the reference sequences of H1N1 and H3N2

challenge viruses in Geneious (version 2021.0.3). IQ-TREE was used to construct the maximum likelihood tree of each gene segment in 1000 bootstrap replicates with the best-fit nucleotide substitution model auto-detected by its curate package – ModelFinder – to further verify the origin of plaque segments (*Nguyen et al., 2015*). The tree files were visualized in FigTree (*Rambaut, 2009*). Based on the origins of the eight gene segments, a genotype was assigned to each plaque isolate. A viral plaque was considered a reassortant if there were one or more gene segment substitutions of both parental H1N1 and H3N2 challenge viruses. A mixed genotype denomination was assigned to those reassortants from plaques that in a given gene segment had complete gene segments from both of the parental viral strains. We labeled the reassortants differently whether they had a substitution with a single gene in a given gene segment (R01–R17) or had more than one (mixed genotypes, M01–M16). The maximum likelihood tree for the sequences of each gene segment is displayed in *Figure 2—source data 1*.

## Identification of within-host variants

The variant calling pipeline has been described previously elsewhere (*Moncla et al., 2020*). In this study, we only identified the H1N1 (A/swine/Minnesota/PAH-618/2011) and H3N2 (A/swine/Minnesota/080470/2015) within-host variants on the gene reads directly sequenced from the BALF samples. Briefly, the trimmed reads were imported and mapped to the reference sequences of the H1N1 and H3N2 challenge viruses in Geneious (version 2021.0.3) (*Kearse et al., 2012*). We excluded the consensus sequences from any IAV segments whose average coverages were less than 100 reads (*Moncla et al., 2020*). The mapped reads were exported in SAM format, sorted by Picard Sortsam (command: java -jar picard.jar SortSam INPUT=input_file.sam OUTPUT=output_file.sam SORT_ORDER=coordinate), and duplicate reads removed using Picard MarkDuplicates (command: java -jar picard.jar MarkDuplicates REMOVE_DUPLICATES=true INPUT=input_file.sam OUTPUT=output_file.sam METRICS_FILE=test.sam.metrics) to avoid PCR bias (http://broadinstitute.github.io/picard/). The files were converted to mpileup files by Samtools (https://github.com/samtools/samtools; *Davies, 2022*) and the viral variants were called by VarScan (https://github.com/dkoboldt/varscan; *Koboldt, 2019*) with the command: samtools mpileup -f reference_file.fasta input_file.sam | java -jar VarScan.v2.3.9.jar pileup2snp >output_file.vcf (*Koboldt et al., 2009*; *Li et al., 2009*). The reported variants were filtered with a minimum depth of 100 reads, the minimum frequency of 1%, mean PHRED quality score of 30, and with the variant detected in both forward and reverse reads by performing the command: java -jar VarScan.v2.3.9.jar filter input_file.vcf `--min-coverage` 100 `--min-avg-qual` 30 `--min-var-freq 0.01` `--min-strands2` 2 `--output-vcf 1>output_file.vcf`. For the samples that contained both challenge viruses, the generated variant report was checked and corrected using a custom Python script to ensure no variants were mistakenly recorded in the report due to nucleotide differences between the challenge viruses, given that mis-mapping of the reads against the reference templates could occur due to the close genetic distance of internal gene segments between H1N1 and H3N2 viruses. For the called SNVs that were located in the same nucleotide sites of H1N1 and H3N2 genomes from the same sample, we checked the original mapping reads and withdraw the false-positive SNVs from the variant report. The final identified variants listed in vcf files were parsed and annotated based on their effect of amino acid changes on reference sequences by the custom Python script. The SNVs identified in H1N1 and H3N2 coding regions were reported based on the H1 and H3 numbering schemes, respectively, including the signal peptide.

## Evolutionary analysis

The within-host evolutionary rates were calculated separately for synonymous, nonsynonymous, and stop-gained mutations at the whole-genome and individual protein level based on methods described previously (*Xue and Bloom, 2020*). The evolutionary rate was calculated for each sample by summing the frequencies of SNVs by their mutation type and divided by available sites and the number of days post-challenge. All BALF samples were collected at 7 days post-contact with the seeder pigs, and the available sites for synonymous, nonsynonymous, and stop-gained mutations were normalized by multiplying the total length of nucleotide sequences for each coding region or all sequenced coding regions combined together (whole genome) by 25, 72, and 3%, respectively. These proportions of sites in IAV genome available for different mutation types were calculated in previous published research (*Xue and Bloom, 2020*), based on counting the proportion of available sites for

each mutation type on the genome of A/Victoria/361/2011 (H3N2) by the Nei and Gojobori method (stop-gained mutations are split from the nonsynonymous mutations) (*Nei and Gojobori, 1986*), and assuming the frequency of transitions is threefold higher than transversions (*Bloom, 2014*; *Pauly et al., 2017*; *Sanjuán et al., 2010*). The efficacy of selection for the IAV populations was estimated by calculating the nucleotide diversity (Pi), synonymous nucleotide diversity (piS), and nonsynonymous nucleotide diversity (piN) at the individual coding region or whole-genome (all sequenced coding regions combined together) level based on the identified within-host variants for each sample by SNPGenie (*Nelson and Hughes, 2015*) (https://github.com/chasewnelson/SNPGenie; *Nelson, 2021*).

To identify the variants located in functional relevant sites, the data of all currently available functional annotations in H1, H3, N1, N2, and the other internal segments were downloaded from the Sequence Feature Variant Types tool available in the Influenza Research Database (*Noronha et al., 2012*). The nonsynonymous variants were annotated if they fell into the annotated region or sites. The annotated functional relevant variants were further categorized based on antiviral drug resistance; determinant of pathogenicity, virulence, and disease progression; host–virus interaction machinery; virus assembly, budding, and release; viral genome transportation, transcription, and replication; viral genome/protein interaction; cross-species transmission and adaption.

The permutation test on shared variant sites was performed at the amino acid level to identify whether the BALF samples from each treatment group shared more polymorphic sites than random chance by the custom Python script adapted from https://github.com/blab/h5n1-cambodia/blob/master/figures/figure-5b-shared-sites-permutation-test.ipynb (*Moncla, 2019*). The permutation test was performed as described before (*Moncla et al., 2020*). Briefly, for each treatment group, we counted the number of variable amino acid sites, n, on each coding region for each sample. We started the permutation test for each group by randomly simulating n variable amino acid sites at each coding region for each sample. Within each iteration, we count the total number of shared polymorphic amino acid sites from all IAV coding regions for each group after removing the amino acid sites shared between pigs from the same room. The null distribution was generated by calculating the total number of polymorphic amino acid sites shared by at least two pigs from different rooms for each treatment group through 100,000 simulations of each gene segment and BALF sample. The p-value was calculated for each treatment group as the total number of iterations in which an equal or larger number of polymorphic sites were shared compared to those observed in actual data, divided by the number of simulations, which was 100,000. Since previously published work indicates that the IAV genome is highly constrained with about 30% of mutations being lethal (*Visher et al., 2016*), we assumed that only 70% of the IAV genome could tolerate mutations. Therefore, we ran the permutation test on both 70 and 100% of each coding region's amino acid sites. All the Python scripts used for evolutionary analysis are available at https://github.com/TorremorellLabUMN/Swine-IAV-within-host-evolution/tree/Main/Script.

## Statistical analysis

Statistical analysis was conducted with the R program version 3.6.2 (*Chambers, 2008*). The percentage of IAV reassortants by treatment groups or vaccination status was compared using a binomial logistic regression model, allowing for overdispersion. The pairwise differences between groups were compared using the chi-squared deviance test, and the p-value was adjusted for multiple comparisons using the Bonferroni–Holm method. This model accounted for the unequal number of pigs from different treatment groups and different number of plaques isolated from each individual pig. The significant results represent the statistical differences at the individual pig level. Kruskal–Wallis rank-sum test was utilized to compare the means of SNV quantity and frequency, nucleotide diversity (Pi), and evolutionary rates by synonymous mutation, nonsynonymous mutation, and stop-gained mutation between treatment groups. The Dunn's test was performed for the pairwise comparisons, and the p-values were adjusted using the Benjamini–Hochberg method. The paired *t*-test was applied to assess the significant differences between the mean piNs and mean piSs within each treatment group for each coding region. The Spearman's rank-order correlation test was performed to evaluate the strength and direction of associations between the percentage of IAV reassortants and co-infection days in pigs. The same statistical analysis also was computed to compare the correlation between HI titers (log2 transformed) and ELISPOT cell counts versus the values of nucleotide divergence, Pi, PiN, and PiS of the HA segment in H1N1 and H3N2 viruses.

## Data availability

The raw sequence reads generated in this study have been deposited in SRA (NCBI) under BioProject accession number PRJNA813974. All the raw datasets and custom Python scripts generated in this study are available in the GitHub repository at https://github.com/TorremorellLabUMN/Swine-IAV-within-host-evolution; *Li, 2022*.

## Acknowledgements

This study was supported by funding from Zoetis. The authors gratefully acknowledge Aaron Rendahl for his help with statistical methodology and the services provided by the University of Minnesota Genomics Center (UMGC) and the Minnesota Supercomputing Institute (MSI).

## Additional information

### Competing interests

Lucina Galina Pantoja: Lucina Galina Pantoja is employed by Zoetis during the time of the study. Micah L Jansen: Micah L Jansen is employed by Zoetis during the time of the study. The other authors declare that no competing interests exist.

### Funding

| Funder | Grant reference number | Author |
| --- | --- | --- |
| University of Minnesota Leman China Scholarship | Graduate Student Fellowship | Chong Li |
| University of Minnesota | | Chong Li Montserrat Torremorell |
| Zoetis | | Montserrat Torremorell Marie R Culhane |

The funders had no role in study design, data collection and interpretation, or the decision to submit the work for publication.

### Author contributions

Chong Li, Conceptualization, Data curation, Formal analysis, Validation, Investigation, Visualization, Methodology, Writing - original draft, Project administration, Writing – review and editing; Marie R Culhane, Montserrat Torremorell, Conceptualization, Resources, Supervision, Funding acquisition, Validation, Writing – review and editing; Declan C Schroeder, Resources, Supervision, Methodology, Writing – review and editing; Maxim C-J Cheeran, Resources, Supervision, Writing – review and editing; Lucina Galina Pantoja, Micah L Jansen, Resources, Funding acquisition, Writing – review and editing

### Author ORCIDs

Chong Li http://orcid.org/0000-0001-6375-0181
Montserrat Torremorell http://orcid.org/0000-0002-9626-6537

### Ethics

All animal and biosafety work that included the collection and handling of samples for this study was approved by the University of Minnesota Institutional Animal Care and Use Committee (Protocol ID: 1712-35407A) and the Institutional Biosafety Committee (Protocol ID: 1508-32918H).

### Decision letter and Author response

Decision letter https://doi.org/10.7554/eLife.78618.sa1
Author response https://doi.org/10.7554/eLife.78618.sa2

## Additional files

### Supplementary files

• Supplementary file 1. Comparison of humoral immune response in pigs whose bronchoalveolar lavage fluid (BALF) samples are available for direct sequencing or plaque purification between treatment groups.

• Supplementary file 2. Number of bronchoalveolar lavage fluid (BALF) samples and number of plaques available for the study.

• Supplementary file 3. All identified functional single-nucleotide variants (SNVs) in the H1N1 virus with annotations.

• Supplementary file 4. All identified functional single-nucleotide variants (SNVs) in the H3N2 virus with annotations.

• Supplementary file 5. Spearman correlation between humoral and cellular immune responses and influenza A virus (IAV) hemagglutinin (HA) nucleotide diversity in individual pigs.

• Supplementary file 6. Clade classification and homology of each gene segment between A/swine/ Minnesota/PAH-618/2011 (H1N1) and A/swine/Minnesota/080470/2015 (H3N2) viruses.

• MDAR checklist

### Data availability

The raw sequence reads generated in this study have been deposited in SRA (NCBI) under Bioproject accession number PRJNA813974. All the raw datasets and custom python scripts generated in this study are available in the GitHub repository: https://github.com/TorremorellLabUMN/Swine-IAV-within-in-host-evolution (copy archived at swh:1:rev:522346731fd666f239122fe9e3dada0e0d3ea141).

The following datasets were generated:

| Author(s) | Year | Dataset title | Dataset URL | Database and Identifier |
|---|---|---|---|---|
| Li C | 2022 | Influenza A virus reassortment and genetic variation in lower respiratory tracts of vaccinated and unvaccinated pigs | https://www.ncbi.nlm.nih.gov/sra/?term=PRJNA813974 | NCBI Sequence Read Archive, PRJNA813974 |
| Li C | 2022 | Swine-IAV-within-host-evolution | https://github.com/TorremorellLabUMN/Swine-IAV-within-host-evolution | GitHub repository, IAV-within-host-evolution |

The following previously published dataset was used:

| Author(s) | Year | Dataset title | Dataset URL | Database and Identifier |
|---|---|---|---|---|
| Li C | 2020 | influenza A virus / Heterologous prime-boost study | https://www.ncbi.nlm.nih.gov/nuccore/?term=MT377710:MT377725[accn] | NCBI GenBank, MT377725 |

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
