## [Editor Report]

Vaccines are a major influenza control strategy in swine but perform sub-optimally and are under-utilized. The manuscript describes a detailed genetic characterization of influenza virus variants in vaccinated versus unvaccinated pigs. The results indicate that viral reassortment, which is an important process yielding new strange influenza viruses of importance to man and animals, may be less common in pigs that have been vaccinated against influenza.

---

## [Decision Letter]

**Decision letter after peer review:**

Thank you for submitting your article "Vaccination decreases the risk of influenza A virus reassortment with a concomitant increase in subgenomic genetic variation in pigs" for consideration by *eLife*. Your article has been reviewed by 3 peer reviewers, including Ron AM Fouchier as the Reviewing Editor and Reviewer #1, and the evaluation has been overseen by Jos van der Meer as the Senior Editor.

Influenza A viruses evolve rapidly in US swine, generating novel reassortants that also occasionally infect humans (e.g., H3N2v). The virus is not well controlled in US swine herds, because of the difficulty of designing vaccines that protect against the full diversity of strains. This research group and others have done prior work examining different prime-boost strategies to improve the effectiveness of swine influenza vaccines. Here, the authors extend that work by examining the potential for vaccination to blunt the evolution of the virus. A strength of the paper is that the challenge was done via a more or less natural route, by comingling seeder pigs infected with H1N1 and H3N2 virus with the vaccinated groups. On the other hand, despite relatively large numbers of animals in the original study, this still does not allow a very robust assessment (statistically) of the impact of vaccination on reassortment generation at the level of individuals pigs.

The study finds that H1N1 and H3N2 viruses readily coinfect pigs, generating a large number of reassortants over a relatively short time period, and possibly fewer reassortants in animals administered the prime-boost regimen. This is an important conclusion, as it seems that vaccination provides an ability to reduce the genetic diversification of swine influenza viruses. This is important also because novel viruses emerging from animal hosts have the ability to spark the next influenza pandemic. However, as it stands, it is unclear if the sample collection was biased and to what extent the results are related to vaccine match or vaccine effectiveness.

Essential revisions:

1. It would be helpful in the introduction to have more background on vaccine use in US swine – ie, how commonly vaccines are used, what types, known efficacy, ongoing challenges, etc. There is a great onus to have a balanced discussion, including vaccine risks and mention of VAERD.

2. The study is using samples generated in a previous study (doi.org/10.1186/s13567-020-00810-z). It is unclear how the samples that were used in the present study were selected from the previous study, which introduces significant concerns about selection bias. The authors should describe the many potential biases that are likely in play.

3. The present study does not discuss the homology between vaccine and challenge strains. Readers have to refer to Table 2 in the parent paper to find the data. Upon closer examination, at least one of the components of every prime-boost treatment had a vaccine with >95% amino acid homology to a challenge virus. Furthermore, it appears that 25% of the sequenced BALF samples were collected from animals that received at least one dose of an autogenous vaccine that was 99.1% homologous to the H3 challenge virus. With estimation of antigenic distance, it is hard to predict exactly how protective each vaccine would have been. It is very hard to know if the prime-boost strategy or similarity between the vaccine strains and challenge virus the is truly responsible for the observed results. To connect the dots between vaccination and evolution it would help to first document how well the vaccines in this trial perform. It is unclear how well the strains in the commercial vaccines match the H1N1 and H3N2 challenge strains, which has a large effect on protection. It is unclear how well the vaccines reduce the attack rate in vaccinated versus unvaccinated animals. Did the attack rate differ between H1 and H3? It is unclear if the vaccines reduce viral shedding in vaccinated versus unvaccinated animals (volume of shedding (Ct) as well as duration of shedding in days). The Results section needs an opening paragraph that describes how well these vaccines protect the animals in general, using multiple measures, before launching into the details of how they impact evolution.

4. Vaccination had limited effect on natural selection of other genetic variants (SNVs). There are many within-host studies showing that immune selection does not occur for influenza during these short active infections. As the Results section on nucleotide variation (L178-L336) is very long, it should be condensed considerably.

5. The definition of "reassortant" in lines 137-139 is confusing. You say 54 were reassortants, but of those 18 were mixed infections. Shouldn't you categorize the mixed infections as their own separate category, not as reassortant? When you refer to "reassortants" in the rest of the paper are you including this mixed infection category? Please define precisely.

6. Given that the proportion of identified reassortants was rather high, perhaps higher than expected from the number of reassortants detected in pigs in real life situations, a discussion about this apparent discrepancy is warranted. It is likely that this is related to lower "fitness" of most reassortants as compared to the ancestral viruses. Were the reassortants generated in this experiment similar to the dominant reassortants commonly identified through surveillance in the field? The authors should discuss the relative fitness of the identified reassortants. Also, given the short time interval between vaccination and seed infection, did you observe any LAIV reassortants? Did you observe any pig-to-pig transmission of reassortants generated in this experiment?

7. What can be said about the 3 super-reassortant pigs? Is there anything distinguishing about them? Viral load, etc.? Is the infection history of the pigs used in these experiments known?

8. The authors conclude that vaccination decreases the risk of reassortment. This may be true at the level of the pooled numbers of analyzed plaques, but is this also true at the level of individual pigs? We see reassortants in 3/4 unvaccinated pigs, in 2/4 single LAIV pigs (with only 2 plaques tested for 1 pig) and in 1/5 prime boost pigs (with one 1 or 3 plaques tested for 2 pigs). Is this a statistically robust conclusion? This statistical analysis should be presented and discussed better.

9. The Results section L96-L128 is the same as in the methods section. It can be deleted in the Results section to save space in favor of the rebuttal to this review.

10. Authors could consider a cartoon that clearly spells out the hypotheses (Vaccination -> immune response -> less coinfection -> less reassortment, less genomic diversity. Vaccination -> immune response -> more selection for new antigenic variants -> more antigenic diversity), perhaps along with the experimental set-up (Figure 1).

---

## [Author Response]

Essential revisions:1. It would be helpful in the introduction to have more background on vaccine use in US swine – ie, how commonly vaccines are used, what types, known efficacy, ongoing challenges, etc. There is a great onus to have a balanced discussion, including vaccine risks and mention of VAERD.

As suggested, we have added several sentences (L73 – 88) to introduce the background information on IAV vaccine usage in the US swine industry and the possible risk of VAERD occurring in pigs after vaccination.

2. The study is using samples generated in a previous study (doi.org/10.1186/s13567-020-00810-z). It is unclear how the samples that were used in the present study were selected from the previous study, which introduces significant concerns about selection bias. The authors should describe the many potential biases that are likely in play.

As suggested, we have amended the detail of how samples were selected for direct sequencing and plaque purifications in lines 155 – 162 in the Results section and lines 578 – 590 in the Materials and methods section. We also discuss in the Discussion section how sample selection could be biasing the results (lines 469 – 485).

3. The present study does not discuss the homology between vaccine and challenge strains. Readers have to refer to Table 2 in the parent paper to find the data. Upon closer examination, at least one of the components of every prime-boost treatment had a vaccine with >95% amino acid homology to a challenge virus. Furthermore, it appears that 25% of the sequenced BALF samples were collected from animals that received at least one dose of an autogenous vaccine that was 99.1% homologous to the H3 challenge virus. With estimation of antigenic distance, it is hard to predict exactly how protective each vaccine would have been. It is very hard to know if the prime-boost strategy or similarity between the vaccine strains and challenge virus the is truly responsible for the observed results. To connect the dots between vaccination and evolution it would help to first document how well the vaccines in this trial perform. It is unclear how well the strains in the commercial vaccines match the H1N1 and H3N2 challenge strains, which has a large effect on protection. It is unclear how well the vaccines reduce the attack rate in vaccinated versus unvaccinated animals. Did the attack rate differ between H1 and H3? It is unclear if the vaccines reduce viral shedding in vaccinated versus unvaccinated animals (volume of shedding (Ct) as well as duration of shedding in days). The Results section needs an opening paragraph that describes how well these vaccines protect the animals in general, using multiple measures, before launching into the details of how they impact evolution.

Thank you for these valuable suggestions. We have rephrased the opening paragraphs in the Results section to document the overall protection efficacy of different prime-boost vaccination protocols, and the difference of infection rates between the H1 and H3 viruses in vaccinated and unvaccinated pigs in lines 139 – 153 and 165 – 167 and discussed it in lines 487 – 493. The information of HA amino acid identity between the vaccine and challenge viruses has also been added in the Material and Methods section in lines 547 – 555.

4. Vaccination had limited effect on natural selection of other genetic variants (SNVs). There are many within-host studies showing that immune selection does not occur for influenza during these short active infections. As the Results section on nucleotide variation (L178-L336) is very long, it should be condensed considerably.

We have removed the result subsection “SNVs were identified at functional sites in both vaccinated and unvaccinated pigs” and the subsection “Limited shared diversity in H1N1 and H3N2 IAV detected in vaccinated pigs”. We significantly condensed the information of these two subsections and combined them in the subsection “Vaccine-induced immunity has limited impact on within-pig HA nucleotide variation of H1N1 and H3N2 viruses”. In addition, we have made multiple arrangements in the other result subsections on nucleotide variations to condense the length of the manuscript.

5. The definition of "reassortant" in lines 137-139 is confusing. You say 54 were reassortants, but of those 18 were mixed infections. Shouldn't you categorize the mixed infections as their own separate category, not as reassortant? When you refer to "reassortants" in the rest of the paper are you including this mixed infection category? Please define precisely.

We define reassortants as any viral isolate that contains gene segments from both parental viruses used in the challenge inoculation. We divided the reassortants in two groups, (a) reassortants R01-R17 which contained a single complete gene sequence in a given gene segment, and (b) reassortants M01-M16 that contained complete gene sequences of both parental viruses in a given gene segment. Therefore, instead of naming the “mixed infection” reassortants, we prefer just to use the word “mixed genotypes” for reassortants with genotypes M01 – M16. When we refer to “reassortants” throughout the paper we include the mixed genotypes as part of the reassortants. We have added the clarifications on the reassortant definition in lines 182 – 184, 668 – 670, 1065 – 1066, and 1076 – 1077.

6. Given that the proportion of identified reassortants was rather high, perhaps higher than expected from the number of reassortants detected in pigs in real life situations, a discussion about this apparent discrepancy is warranted. It is likely that this is related to lower "fitness" of most reassortants as compared to the ancestral viruses. Were the reassortants generated in this experiment similar to the dominant reassortants commonly identified through surveillance in the field? The authors should discuss the relative fitness of the identified reassortants. Also, given the short time interval between vaccination and seed infection, did you observe any LAIV reassortants? Did you observe any pig-to-pig transmission of reassortants generated in this experiment?

The reviewer raises a good point regarding the much higher proportion of reassortants identified in this study than the pigs from “real-life” situations. One factor that may contribute to this is that all our reassortants are from lungs (BALF samples) and that the transmissibility of IAV reassortants may be influenced by tissue tropism (Zhang et al., 2018). In Zhang et al., reassortant viruses were more abundant in the middle and lower respiratory tract rather than the upper respiratory tract in the nasal cavities (Zhang et al., 2018). This may be related to the high temperature and affluent presence of various types of sialic acid receptors present in the lower respiratory tract (Nelli et al., 2010; Zhang et al., 2018; Graaf & Fouchier., 2014). Thus, is possible that not all the reassortants that are generated in the pigs are likely to transmit which could explain the differences observed under field conditions. In addition, other host and environmental factors could also affect the fitness and circulation of IAV reassortants (Lowen., 2018). We have added this discussion in the manuscript in lines 406 – 413 and 419 – 420.

Regarding the question on whether we observed any reassortant related to the LAIV vaccine, the answer is no, we didn’t. We collected nasal swabs every day from all LAIV vaccinated pigs after LAIV vaccine administration and performed the RRT-PCR which targets the LAIV vaccine strains to evaluate LAIV shedding. The shedding period for pigs receiving the LAIV vaccine was from 1 to 9 days post vaccination. At 10 days post LAIV vaccination, there was no LAIV detected in the vaccinated pigs. Then, there were five weeks between pigs receiving the LAIV vaccine and contact with the seeder pigs, so LAIV vaccinated pigs did not have the vaccine virus by the time they were mixed with the seeder pigs. Therefore, the conditions to generate reassortant viruses for the LAIV group were not met and as a result we didn’t detect any reassortant with sequences from the LAIV vaccines. We have added information on this in the Results section in lines 188 – 189.

Our study was not set up to assess transmissibility of the reassortants. However, within the limitations of our study, we did not find convincing evidence of pig-to-pig transmission of the reassortants generated in this study. Even though we found specific reassortants (R01, R10, M01, M06) in the lungs of multiple pigs, only M01 reassortants were detected in pigs that were co-housed in the same room. However, whether the M01 reassortant was the result of pig-to-pig transmission or it was generated in two pigs independently is hard to tell. Additional studies are needed to better understand the factors that contribute to the transmissibility of the reassortant viruses.

In terms of understanding how the reassortants generated in this study relate to the dominant viruses circulating in the field, the hemagglutinin (HA) of the challenge viruses and generated reassortants in this study belong to major circulating lineages of the swine influenza viruses in US farms. This information is based on the influenza A virus swine surveillance quarterly reports (USDA-APHIS). Except for the PA and NP genes, the internal genes of the two challenge viruses belong to the same lineages (based on the IAV evolutionary lineage classification tool – OctoFLU). Therefore, many reassortants generated in this study kept the same genotype with one of their parental viruses. We have downloaded all the available whole genome data of 325 H1N1 and 434 H3N2 IAV strains isolated from US swine populations since 2017 from the Influenza Research Database (IRD) and performed the analysis to identify their genotypes. We found that about 43.4% (141/325) of analyzed H1N1 viruses have the same genotype as the R13, R14, and M14 viruses generated in this study. Among the H3N2 reassortants generated in this study, the R02 and R16 viruses are most commonly identified in the field. About 9.2% (40/434) of analyzed H3N2 viruses have the same genotype as these viruses. No H3N2 reassortants in this study have the same genotype as the dominant H3N2 viruses recently described in US pig populations. That is because the N2 gene of the H3N2 parental virus belongs to the N2-2002A lineage, which is not a major N2 lineage. If this H3N2 parental virus replaces an N2-2002B N2 gene, it will have the same genotype as about 58.5% (254/434) of the analyzed H3N2 viruses.

7. What can be said about the 3 super-reassortant pigs? Is there anything distinguishing about them? Viral load, etc.? Is the infection history of the pigs used in these experiments known?

The reviewer raises a very important observation, and we are also intrigued by the finding that some pigs appeared to be “super-reassortant pigs”. The three super-reassortant pigs didn’t exhibit notable clinical differences, except for the fact that these three pigs had longer simultaneous shedding duration of both challenge viruses (figure 3 – source data 1). These three super-reassortant pigs (one SINGLE LAIV pig and two NO VAC pigs), like the other NO VAC and SINGLE LAIV pigs, had no detectable humoral response specific to the H1N1 and H3N2 challenge viruses prior to the challenge. The IAV infection dynamics of pigs enrolled in the analysis of reassortant identifications is shown in Author response image 1. Even though we found that two of three super-reassortant pigs had higher virus shedding quantity than most pigs after 5 dpc, the difference in virus load was not remarkable, especially in BALF samples collected at 7 dpc. Besides, all pigs showed mild gross lung lesions (0.5% – 10%). Lastly, the three super-reassortant pigs were housed in three different rooms which indicates that there wasn’t a “room effect”. Whether there were other host factors responsible for creating super reassortant pigs is unknown at this time.

**Author response image 1. sa2fig1:** Dynamics of overall influenza shedding in pigs that were part of the reassortant analysis. The quantity of virus shedding in nasal swabs and BALF samples were measured by influenza matrix gene real-time PCR (A) and virus titrations (B). The individual pigs are represented by different colors and their treatment groups are shown in different shapes. The dark red line indicates the median virus quantity shed by 13 analyzed pigs at each time point. The three super-reassortant pigs are labeled with the red symbol borders and also presented in the figure legends.

8. The authors conclude that vaccination decreases the risk of reassortment. This may be true at the level of the pooled numbers of analyzed plaques, but is this also true at the level of individual pigs? We see reassortants in 3/4 unvaccinated pigs, in 2/4 single LAIV pigs (with only 2 plaques tested for 1 pig) and in 1/5 prime boost pigs (with one 1 or 3 plaques tested for 2 pigs). Is this a statistically robust conclusion? This statistical analysis should be presented and discussed better.

Thank you for bringing up this valuable point. To compare the percent of influenza reassortant plaques between pigs from different treatment groups, we applied a binomial logistic regression model, allowing for overdispersion, and tested all pairwise differences between treatments using the chi-squared deviance test, and adjusted the p-values for multiple comparisons using the Bonferroni-Holm method. This model accounted for the unequal number of pigs from different treatment groups and different number of plaques isolated from each individual pig. Therefore, the significant results represent the statistical differences at the individual pig level. However, our statistical model is based on the asymptotic theory (ie, what happens when the sample size gets large), and we agree with the reviewer that the results are still based on a limited number of pigs. Therefore, we provided more detail in the statistical analysis and addressed the result and discussion in lines 199 – 201, 497 – 503, 763 – 766, and 1077 – 1081.

9. The Results section L96-L128 is the same as in the methods section. It can be deleted in the Results section to save space in favor of the rebuttal to this review.

Agreed. We have rephrased and condensed the texts of this part to save the space for the newly added paragraphs, which provided the background information about the protection of multiple vaccine combinations in pigs against influenza infections and illustrated the hypothesis of this study.

10. Authors could consider a cartoon that clearly spells out the hypotheses (Vaccination -> immune response -> less coinfection -> less reassortment, less genomic diversity. Vaccination -> immune response -> more selection for new antigenic variants -> more antigenic diversity), perhaps along with the experimental set-up (Figure 1).

As suggested, we added the cartoon in figure 1 to show the hypothesis of this research. The hypothesis is also illustrated in Results section at lines 168 – 172.